# The genome-wide multi-layered architecture of chromosome pairing in early *Drosophila* embryos

Jelena Erceg[1,10], Jumana AlHaj Abed [1,10], Anton Goloborodko [2,10], Bryan R. Lajoie[3,6], Geoffrey Fudenberg [2,7], Nezar Abdennur[2], Maxim Imakaev [2], Ruth B. McCole[1], Son C. Nguyen [1,8], Wren Saylor [1], Eric F. Joyce [1,8], T. Niroshini Senaratne[1,9], Mohammed A. Hannan[1], Guy Nir [1], Job Dekker [3], Leonid A. Mirny [2,4] & C.-ting Wu[1,5]

Genome organization involves *cis* and *trans* chromosomal interactions, both implicated in gene regulation, development, and disease. Here, we focus on *trans* interactions in *Drosophila*, where homologous chromosomes are paired in somatic cells from embryogenesis through adulthood. We first address long-standing questions regarding the structure of embryonic homolog pairing and, to this end, develop a haplotype-resolved Hi-C approach to minimize homolog misassignment and thus robustly distinguish *trans*-homolog from *cis* contacts. This computational approach, which we call Ohm, reveals pairing to be surprisingly structured genome-wide, with *trans*-homolog domains, compartments, and interaction peaks, many coinciding with analogous *cis* features. We also find a significant genome-wide correlation between pairing, transcription during zygotic genome activation, and binding of the pioneer factor Zelda. Our findings reveal a complex, highly structured organization underlying homolog pairing, first discovered a century ago in *Drosophila*. Finally, we demonstrate the versatility of our haplotype-resolved approach by applying it to mammalian embryos.

---

[1] Department of Genetics, Harvard Medical School, Boston, MA 02115, USA. [2] Institute for Medical Engineering and Science, Massachusetts Institute of Technology (MIT), Cambridge, MA 02139, USA. [3] Howard Hughes Medical Institute and Program in Systems Biology, Department of Biochemistry and Molecular Pharmacology, University of Massachusetts Medical School, Worcester, MA 01605-0103, USA. [4] Department of Physics, Massachusetts Institute of Technology (MIT), Cambridge, MA 02139, USA. [5] Wyss Institute for Biologically Inspired Engineering, Harvard University, Boston, MA 02115, USA. [6] Present address: Illumina, San Diego, CA, USA. [7] Present address: Gladstone Institutes of Data Science and Biotechnology, San Francisco, CA 94158, USA. [8] Present address: Department of Genetics, Penn Epigenetics Institute, Perelman School of Medicine, University of Pennsylvania, Philadelphia, PA 19104-6145, USA. [9] Present address: Department of Pathology and Laboratory Medicine, David Geffen School of Medicine, University of California, Los Angeles, CA 90095, USA. [10] These authors contributed equally: Jelena Erceg, Jumana AlHaj Abed, Anton Goloborodko. Correspondence and requests for materials should be addressed to L.A.M. (email: leonid@mit.edu) or to C.-t.W. (email: twu@genetics.med.harvard.edu)

Although chromosomes are organized within the nucleus into distinct territories, they nevertheless come into contact *in trans*. In diploid organisms, including mammals, such *trans* contacts may involve specific interactions, such as pairing, between the homologous maternal and paternal chromosomes. Although best known in meiotic cells, homolog pairing can also occur in somatic cells and influence gene expression via phenomena such as transvection (reviewed by[1–3]). In *Drosophila*, somatic pairing increases extensively from early embryogenesis to adulthood, making this organism an ideal system for studying *trans* interactions (reviewed by[1–3]). What, for example, is the structure of homolog pairing in early embryos? Pairing may encompass many forms, ranging from a well-aligned juxtaposition of homologs (railroad track), to a loose, laissez-faire association of homologous regions, or even an apposition of highly disordered structures; for instance, pairing in mammalian systems can manifest as a nonrandom, approximate co-localization of homologous chromosomal regions (reviewed by[1,2]).

Recently, high- and super-resolution[4–7] imaging of several genomic loci has suggested that pairing can involve distinct chromosomal domains as well as more closely merged regions, while simulations of pairing[8] via integration of the lamina-DamID data and the Hi-C data have predicted correlations between pairing and epigenetic domains. Furthermore, a transgene-based study of transvection has proposed a domain-based mode for pairing[9]. These studies are consistent with long-standing discussions about the establishment, maintenance, and stability of pairing as well as the relationship between pairing and transcription (reviewed by[1–3,10,11]). Nevertheless, questions regarding the extent and structure of pairing remain. More recently, Hi-C-derived ligation products with overlapping sequences have been used to infer short-range interactions between homologs or sister chromatids in a tetraploid Kc$_{167}$ cell line and thus a correlation between short-range interactions and active regions[12].

Here, we asked how pairing initiates in early *Drosophila* embryos, when maternal and paternal genomes first meet. Although the imaging of individual loci has documented the initiation of pairing during early embryogenesis[13–17], the global structure of pairing and its genome-wide relationship to fundamental developmental programs has remained elusive. To that end, we took advantage of single nucleotide variants (SNVs) to enable haplotype-resolved Hi-C. Haplotype-resolved Hi-C has previously been used to study *cis* interactions in mammalian systems[18–29] as well as *trans*-homolog interactions in yeast[30] and, here, we developed a method (Ohm) to much more confidently identify those *trans* interactions that occur between homologous chromosomes. Importantly, our approach allowed us to detect both short- and long-range interactions, that is, interactions spanning genomic distances as small as a few kilobases to several megabases. Excitingly, haplotype-resolved Hi-C as well as Ohm can be applied to any organism whose genome sequence has been haplotype-resolved, and we demonstrate their generality by applying them also to studies[25,26] of early mammalian embryos.

Strikingly, our Ohm approach reveals that homolog pairing is extensive along entire lengths of chromosomes at short- as well as long-range distances and is highly structured, with *trans*-homolog domains, domain boundaries, compartments, and interaction peaks. We reveal the coincidence between *trans*-homolog features and the positions and sizes of analogous *cis* features and then provide a three-axis framework of precision, proximity, and continuity to accommodate all types of inter-chromosomal interactions, including homolog pairing. These findings complement those of our companion study (AlHaj Abed, Erceg, Goloborodko et al.[31]), which, by generating and then using a *Drosophila* hybrid cell line with a high degree of pairing, enabled a much more in-depth analysis of the structure of pairing. In particular, this other study observed at least two forms of pairing and correlations between these forms and the active or repressed state of chromatin. In our current study, we also explore the relationship between pairing and the developmental programs in play in the early embryo during the initial stages of pairing. We find that pairing correlates with the opening of chromatin mediated by the pioneer factor Zelda during zygotic genome activation.

## Results

**Haplotype-resolved Hi-C for profiling embryonic pairing**. We focused our Hi-C analysis on the window of *Drosophila* embryogenesis extending from 2–4 h after egg laying (AEL), when zygotic gene transcription is activated and the embryo transitions from a syncytial blastoderm into a multicellular state[32]. It is during this window that maternal and paternal homologs come together for the first time[13–15], and the rapid early cleavage cycles give way to the longer 13th and 14th cycles, thus minimizing the contribution of mitotic genome organization to our Hi-C signal[33]. To generate the embryos for our study, we mated two divergent *Drosophila* Genetic Reference Panel lines, 057 and 439 (Fig. 1a, b), as a high density of SNVs is required for assaying homolog pairing via genomic methods. Indeed, strains 057 and 439 differ with an average SNV frequency of ~5.5 per kilobase (kb), except on chromosome 4 (1.0 SNV/kb) (see Supplementary Note 1). This frequency was adjusted to ~5.1 SNVs/kb (0.01 for chr4) when we re-sequenced both lines and reconstructed an F1 diploid genome using only high-confidence homozygous SNVs (Supplementary Fig. 1 and Supplementary Table 1, see Methods); this diploid genome assembly ensured the most stringent haplotype-resolved mapping of Hi-C products. Importantly, we confirmed that hybrid embryos achieved levels of homolog pairing consistent with those observed in other studies[13–17] using fluorescent in situ hybridization (FISH) to assess pairing (Fig. 1c) at heterochromatic (16.0% ± 5.7–62.4% ± 3.6) and euchromatic (2.0% ± 1.5–22.0% ± 2.4) loci across different chromosomes (Fig. 1d, e). In addition, pairing increased to expected levels in 057/439 embryos as they aged (Supplementary Fig. 2a).

We performed Hi-C and recovered 513 million Hi-C products, which after mapping to the reference genome, yielded ~56% unique read pairs (Supplementary Table 2). When assembled into a map and analyzed at 1 kb resolution without regard to parental origin of the mapped fragments, these reads revealed features that are routinely seen in non-haplotype-resolved Hi-C maps[33,34]—a central *cis* diagonal representing short-distance interactions in *cis* (*cis* read pairs) as well as signatures for compartments, domains (also called contact domains[19] and topologically associated domains, or TADs[35,36], hereafter referred to as domains), and interaction peaks—thus confirming the quality of our Hi-C data (Supplementary Fig. 2b). Presence of these interphase hallmarks indicates that the vast majority of the cells are in interphase rather than mitosis, where such features are absent[33].

**Ohm-directed distinction of *trans*-homolog from *cis* contacts**. Using our newly generated haplotype-resolved F1 diploid genome to determine the parental origin of our reads, we found that 5.8% of all mappable read pairs appeared to represent *trans* (*trans*-homolog as well as *trans*-heterolog) contacts and that, of these, 36.1% indicated contacts between homologous chromosomes (*trans*-homolog read pairs) (Supplementary Table 3). Encouraged by these numbers, we next determined the quality of the purported *trans*-homolog reads. For example, errors in reconstruction of the F1 diploid genome or sequencing of the Hi-C products can lead to misassignment of *cis* read pairs as *trans*-homolog read pairs (Fig. 2a) and, given the greater number of *cis* as versus *trans*-homolog read pairs, even a small rate of misassignment of *cis* read pairs could be confounding. Thus, to assess the potential magnitude of homolog

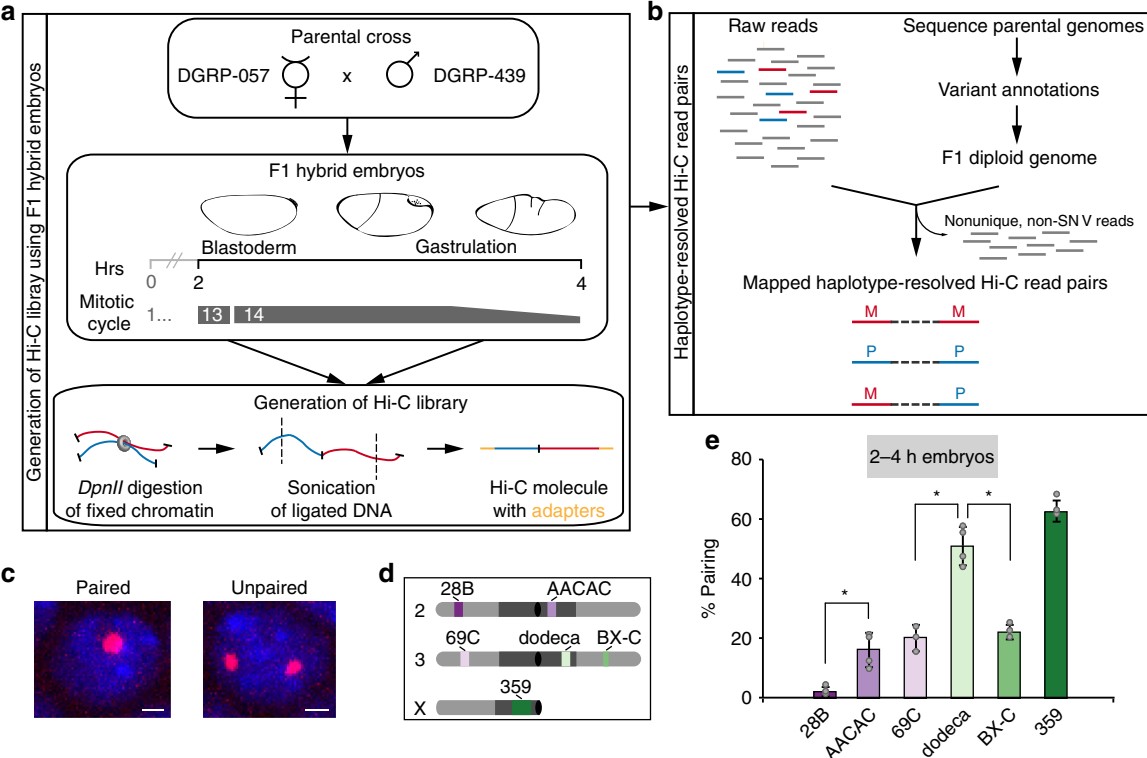

**Fig. 1** Haplotype-resolved Hi-C to characterize embryonic homolog pairing. **a** Generation of Hi-C libraries using 2–4 h hybrid embryos and **b** haplotype-resolved mapping. **c** Homolog pairing as assessed by FISH. Nuclei are considered paired nuclei when FISH signals are ≤0.8 μm (center-to-center distance) apart. Bar = 1 μm. **d** Location of FISH targets (heterochromatin, dark gray; euchromatin, light gray; centromere, black circle). **e** Percentage of nuclei showing paired loci in 2–4 h embryos (error bars, standard deviation of at least three replicates; n ≥ 100 nuclei/replicate; *P < 0.0001, Fisher's two-tailed exact). Source data are provided as a Source Data file

misassignment, we sought a signature for it in our Hi-C dataset, focusing on read pairs where the separation along the genome (genomic separation) of the two reads of a read pair is <1 kb (Fig. 2b, shaded area). We chose to examine these read pairs because they are enriched in non-informative byproducts of the Hi-C protocol[37], the most abundant being unligated pieces of DNA, known as dangling ends (Fig. 2b, hatched area and Supplementary Fig. 3, see Methods). Dangling ends have genomic separations of <1 kb, are "inwardly" orientated, and 100–1,000 fold more abundant than read pairs of other orientations at the same separations (Supplementary Fig. 4a) and, since they are exclusively *cis* read pairs, their enrichment among *trans*-homolog read pairs can only arise via homolog misassignment (HM in figures and Methods). As such, the enrichment of short-distance (<1 kb) *trans*-homolog pairs can serve as a signature of homolog misassignment and be used to assess the overall quality of our homolog assignment (Methods). Consistent with this hypothesis, this signature of homolog misassignment almost disappeared when we increased the stringency of our mapping by requiring at least two SNVs per read (Fig. 2c and Supplementary Fig. 4b, c, probability of homolog misassignment $P_{HM}$ = 0.01%). In contrast, homolog misassignment increased when we used the less accurate DGRP SNV annotations (Supplementary Fig. 4d, $P_{HM}$ = 2.40%). This approach, which we call Oversight of homolog misassignment (Ohm), allowed us to estimate the probability of homolog misassignment as only ~0.17% when requiring at least 1 SNV per read (Fig. 2d, see Methods). Finally, this analysis indicated that homolog misassignment introduced less than 5% of erroneous *trans*-homolog contacts at separations above 1 kb (Fig. 2e). Having gained confidence that contamination contributes only a minor portion of purported *trans*-homolog read pairs, we observed that *trans*-homolog contacts at

genomic separations of 1 kb are >100-fold more frequent than those at genomic separations of >1 Mb, and >500 fold more frequent than contacts between regions on different chromosomes (*trans*-heterolog, Fig. 2b). Thus, *trans*-homolog contacts constitute a major fraction of *trans*-chromosomal interactions.

**Homolog pairing in *Drosophila* embryos is highly structured**. We next generated a Hi-C map using ≥ 1 SNV per read and filtering our data to exclude contacts below 3 kb of genomic separation in order to substantially reduce contamination by Hi-C byproducts. The resulting map was striking in a number of ways (Supplementary Fig. 2b). In addition to displaying the expected central *cis* diagonal, it revealed prominent "*trans*-homolog" diagonals that extended across the entire length of each chromosome (Fig. 3a, black arrows). This observation demonstrated that interactions between homologs, including those within a few kilobases and up to several megabases, extend genome-wide in early embryos at the time when pairing is being initiated.

The *trans*-homolog diagonals suggest that homologs are relatively well-aligned when coming into contact, consistent with a railroad track structure (Fig. 3b). This indication, however, does not preclude less aligned forms of pairing, such as the paired domains observed at a number of genomic regions[4–7]. For example, Fig. 2b also shows that extensive *trans*-homolog interactions may occur between non-allelic regions corresponding to genomic separations of hundreds of kilobases or more, albeit at much reduced frequencies. We would expect such interactions to appear as signals at varying distances off the *trans*-homolog diagonals, leaving open the possibility of other structures, such as laissez-faire (Fig. 3c) and highly disordered pairing (Fig. 3d).

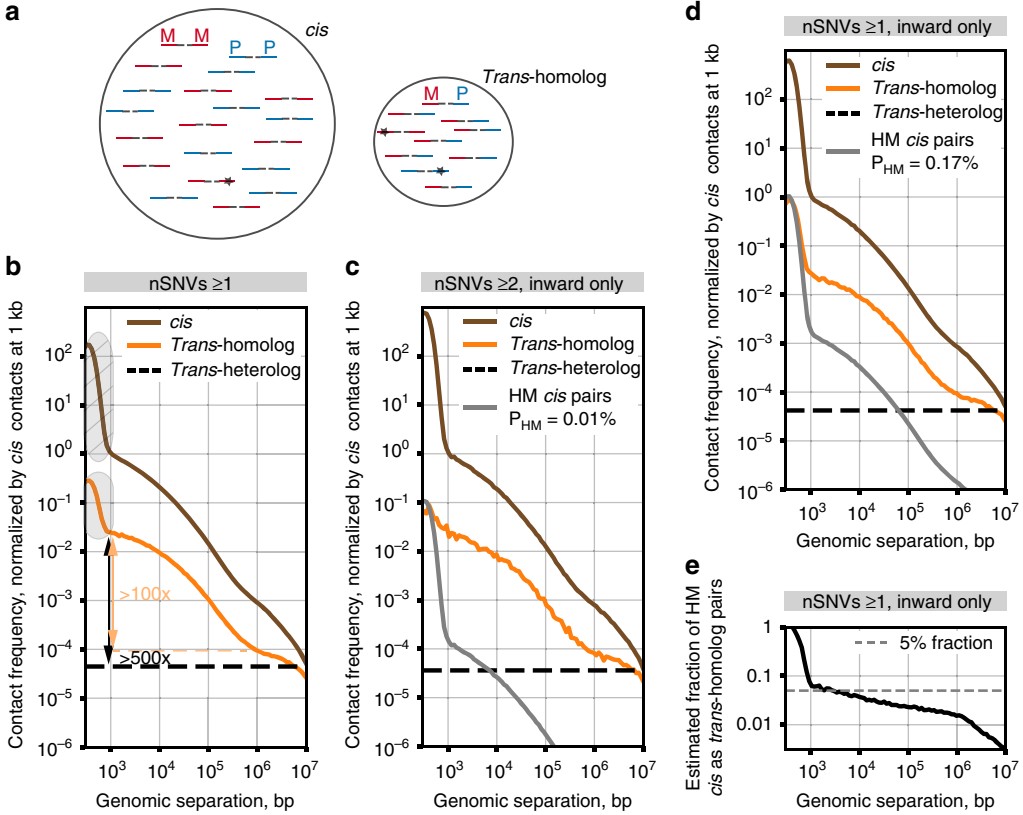

**Fig. 2** Strategy for robust distinction of *trans*-homolog from *cis* contacts. **a** Homolog misassignment can arise from sequencing errors, false SNVs, or sample heterogeneity (lines, Hi-C molecules; red, maternal fragment (M); blue, paternal fragment (P); stars, errors). **b** Contact frequency plotted against genome separation using ≥ 1 SNV per read. Arrows, change in contact frequency between short-range *trans*-homolog contacts and either long-range (>1 Mb) *trans*-homolog (orange) or *trans*-heterolog (black) contacts. Shaded area over the *trans*-homolog curve, enrichment of contact frequency due to homolog-misassigned "dangling ends" (shaded area over *cis* curve). **c** Contact frequency for inward only read pairs with at least two SNVs per read, with no sequence mismatches allowed ($P_{HM} = 0.01\%$). **d** Contact frequency for inward only read pairs with at least one SNV per read ($P_{HM} = 0.17\%$). **e** Fraction of homolog-misassigned *cis* contacts among *trans*-homolog pairs as a function of genomic separation. **b–d** Contact frequencies for chromosomes 2 and 3 normalized by *cis* contact frequency at 1 kb. Dashed black line, average *trans*-heterolog contact frequency; HM, homolog misassignment; $P_{Hm}$, probability of homolog misassignment

That pairing may assume different forms in *Drosophila* has long been considered (reviewed by[1–3]), with recent studies suggesting that the state of pairing is a fine balance between pairing and anti-pairing factors (reviewed by[2]). Here, we propose a framework in which the different aspects of pairing can be placed along the continuum of three axes representing (i) the precision with which homologous regions are aligned (*x* axis), (ii) the proximity with which homologs are held together (*y* axis), and (iii) the continuity or degree to which a particular state of pairing extends uninterrupted (*z* axis), with the continuum along any axis potentially reflecting different forms of pairing and/or the maturation (or degradation) of pairing over time (Fig. 3e). In this context, high values along all three axes would produce railroad track pairing, intermediate values would approximate a laissez-faire mode of pairing, and minimum values would encompass interactions ranging from extreme laissez-faire to disorder. This framework can also be used to describe *trans* interactions between heterologous chromosomes, particularly prominent examples being those arising from the Rabl polarization of centromeres and telomeres, appearing in some Hi-C maps, including ours, as contacts along non-homologous arms (e.g. 2L and 2R, as well as 3L and 3R)[33,34]; the Rabl configuration may facilitate homolog pairing by reducing search space within the nucleus (Fig. 3f[13–15]). Note, however, that *trans*-homolog contacts were tighter and/or more frequent than *trans*-heterolog contacts (Fig. 3g, h), emphasizing the prevalence of homolog pairing in *Drosophila*.

We next asked whether our haplotype-resolved maps could further clarify the molecular structure of paired homologs as well as elucidate how *trans*-homolog interactions are integrated with the *cis* interactions that shape the 3D organization of the genome. Remarkably, we observed *trans*-homolog domains, domain boundaries, compartments, and interaction peaks resembling analogous features in *cis* maps (Fig. 3i–k and Supplementary Fig. 5a–e). In fact, 54% of the boundaries of *trans*-homolog domains overlapped a boundary of a *cis* domain (averaged over two homologs, Supplementary Fig. 5f). While this value is lower than the upper bound of 72% for *trans*-homolog boundaries shared between replicate haplotype-resolved Hi-C datasets, it is nevertheless significantly above the 35% overlap expected at random ($P < 10^{-10}$, *t*-test). Note that, as we obtained 92.3% overlap of domain boundaries between replicates of non-haplotype-resolved Hi-C datasets, it is likely that the overlap values for haplotype-resolved datasets reflect the relatively low number of haplotype-resolved reads we obtained.

The similarity between *trans*-homolog and *cis* contact maps is further highlighted through comparisons with maternal- and paternal-specific Hi-C domains, which have been observed in other systems to be concordant[21]; in our dataset, we observed 73.9% concordance between *cis* domain boundaries of homologous chromosomes (Supplementary Fig. 5f; also Fig. 3l and Supplementary Fig. 5g, h), wherein concordance between two replicates was 73.2% (see Methods). *Trans*-homolog maps do,

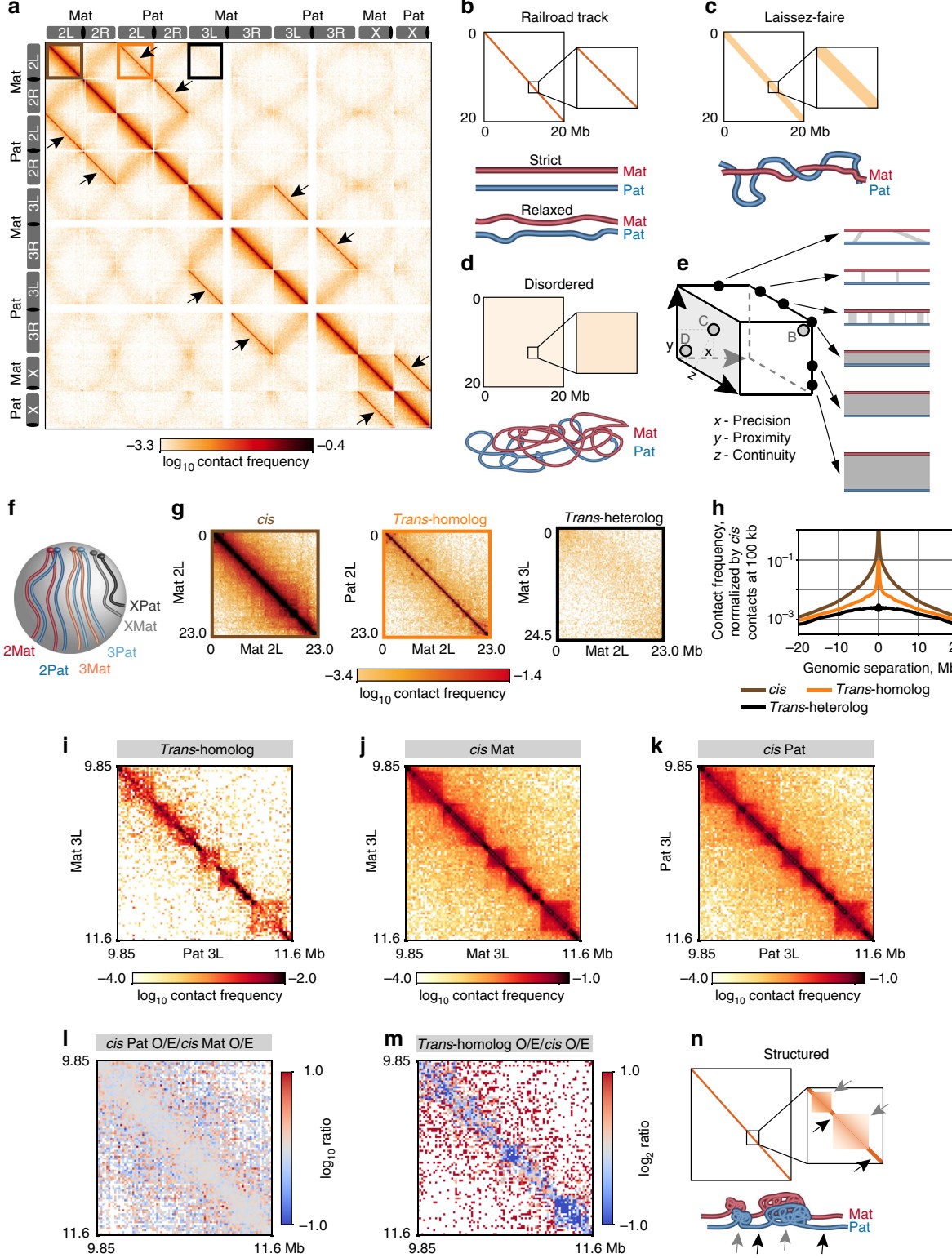

**Fig. 3** Highly structured homolog pairing resembles features of *cis*-organization. **a** Haplotype-resolved Hi-C map of F1 hybrid embryos. Arrows, *trans*-homolog diagonals. **b–d** Homologs juxtaposed in a **b** railroad track fashion, **c** a loose, laissez-faire mode, or **d** a highly disordered structure. **e** Homolog pairing may encompass a range of structures defined by precision, proximity, and continuity. **f** Rabl positioning of chromosomes. The **g** zoomed-in maps of chromosomal arms (color coded boxes in a; L, left; R, right) and **h** respective contact frequencies as a function of genomic distance (see Methods). **i** *trans*-homolog, **j** *cis* maternal, and **k** *cis* paternal maps of matching regions on chr3L. **l** Ratio of *cis* Pat/*cis* Mat Hi-C maps indicates the *cis* contact patterns of two homologs are highly consistent. **m** The ratio of *trans*-homolog/average *cis* maps suggests that pairing resembles *cis* contacts, albeit with lower interactions in some regions (dark blue). **n** Homolog pairing displays highly structured *trans*-domains (black) and *trans*-boundaries (gray), reflecting *cis*-organization of homologs

however, differ from some *cis*-defined features (Fig. 3m and Supplementary Fig. 5i, dark blue; see also[31]). Taken together, our findings indicate that homolog pairing goes far beyond simple genome-wide alignments of homologous pairs and includes surprisingly structured *trans*-homolog domains, boundaries, and specific interaction peaks that frequently correspond to *cis* features (Fig. 3n).

**Homolog pairing is associated with zygotic genome activation.** Spurred on by the rich history of transvection (reviewed by[1–3]) we asked how broadly pairing might be correlated genome-wide with fundamental developmental events such as the binding of major transcription factors in the early *Drosophila* embryo. We focused our attention on arguably the four most prominently studied maternally-contributed factors, involved in zygotic genome activation, in particular key developmental events such as the opening of chromatin, initiation of transcription, and embryonic pattern formation: the pioneer factor Zelda (Zld), which mediates early chromatin accessibility[38,39]; Bicoid (Bcd)[40] and Dorsal (Dl)[41], which are involved in patterning the anterior-posterior (AP) and dorsal-ventral (DV) axis, respectively; and GAGA factor (GAF)[42], which may facilitate transcription in later stages. We correlated the binding of these factors, as revealed by ChIP-seq datasets[40,41,43], to local variations in the degree of pairing as characterized by the pairing score (PS), defined as the $\log_2$ *trans*-homolog contact frequency within a 28 kb window along the diagonal of Hi-C maps (Methods). Excitingly, regions associated with elevated PS values were significantly enriched for the binding of Zld and Bcd ($P < 10^{-10}$, Spearman; Fig. 4a) as compared to controls in which we randomized 200 kb chunks of PS values (Fig. 4b) or 200 kb chunks of binding profiles (Supplementary Fig. 6a). No correlation was observed, however, between PS values and the binding of GAF or Dl ($P = 0.139$, $P = 0.029$, Spearman, respectively; Fig. 4b). Then, using partial correlation analyses, we examined whether these correlations would be affected upon removal of the contribution of a third feature. For instance, how might the correlation between the binding of Bcd and PS values be affected by the binding of Zld? We found that Zld binding contributes significantly, as Spearman's correlation coefficient dropped from 0.155 ($P = 2.1 \times 10^{-65}$; Fig. 4b) to 0.044 ($P = 1.3 \times 10^{-6}$; Supplementary Fig. 6b), when controlled for Zld binding. Interestingly, Spearman's correlation coefficient between the binding of Dl and PS values decreased from $-0.020$ ($P = 0.029$; Fig. 4b) to $-0.157$ ($P = 3.0 \times 10^{-67}$; Supplementary Fig. 6b) when controlled for the contribution of Zld. These findings are consistent with a role of Zld in Bcd- as well as Dl-dependent binding and gene activation[40,41]. Interestingly, we observed that genes associated with both AP and DV patterning[44,45] are similarly associated with higher PS values (Supplementary Fig. 6c). Moreover, loss of Bcd binding in Zld-depleted embryos[40] was significantly enriched at regions otherwise associated with high PS values ($P < 1.21 \times 10^{-68}$, Mood's median test; Supplementary Fig. 6d, e). In sum, our data suggest that Zld-mediated early opening of chromatin, including the binding of Bcd, is strongly correlated with pairing in embryos.

As previous studies suggested that regions with high Zld occupancy were more dependent on Zld for establishing domain boundaries[33], we asked whether Zld may play a role in domains that are established via *trans*-homolog interactions. Several observations supported our hypothesis. First, visual inspection of our Hi-C revealed that *trans*-homolog domain boundaries are associated with high Zld occupancy and, moreover, are associated with values of PS that are higher than those of the interior of domains (Fig. 4a and Supplementary Fig. 7a, $P < 10^{-10}$, Mood's median test). Interestingly, 25% of the top Zld-occupied regions have high PS values ($P < 10^{-10}$, Mood's median test, Fig. 4c).

Second, using Hi-C data from Zld-depleted embryos[33], we observed a decrease of boundary strength at regions that would otherwise be associated with high PS values and strong Zld binding (Fig. 4d and Supplementary Fig. 7b, $P < 10^{-10}$, Spearman). Third, we found that Zld depletion trended strongly toward reduced pairing, with the reduction being significant at four out of six loci assayed ($P < 2.56 \times 10^{-3}$, Fisher's two-tailed exact; Supplementary Fig. 7c, d). Finally, we also observed strong correlations between PS values and features known to be enriched at domain boundaries, such as RNA Pol II[33], housekeeping genes[33,46], and nascent zygotic gene products[47] (Supplementary Fig. 6c, 7e). Thus, Zld-mediated chromatin accessibility and transcription appear correlated with the establishment of homolog pairing in the early embryo (Fig. 4e), reinforcing the notion that the establishment of homolog pairing may bear a relationship to genome function during key developmental events. For example, Zld-mediated opening of genomic regions may facilitate homologous sequences finding each other when pairing is initiated in early embryogenesis. Whether Zelda plays a direct or indirect role in initiating pairing, however, will require additional study, as will the interplay between Zld and other factors involved in pairing and other forms of chromosome organization.

**Haplotype-resolved application of Ohm to mammalian embryos.** One of the most exciting developments in recent years has been the growing indication that homolog pairing is not special to *Drosophila*, that it may be a general property in many organisms, including mammals (reviewed by[1,2]). Thus, in order to both demonstrate the applicability of Ohm as well as assess the state of pairing in mammals, we applied our haplotype-resolved Ohm approach to mammalian embryos. As pairing in mammals is generally transient and localized, for example, during DNA repair, V(D)J recombination, imprinting, or X-inactivation (reviewed by[1,2]), these analyses would, at the least, also serve as a control for observations in *Drosophila*, where pairing is far more extensive. In particular, we considered two deeply sequenced Hi-C datasets from Du et al.[25] and Ke et al.[26] representing gametes and F1 hybrid mouse embryos bearing a high frequency of SNVs (1 autosomal SNV per ~130 bp or ~490 bp, respectively). Reasoning that the haploid nature of sperm and oocytes precludes homolog pairing, we focused first on these cell types to determine the level of stringency that would be necessary to remove any apparent *trans*-homolog read pairs that would necessarily have resulted from homolog misassignment (Supplementary Fig. 8a). That level of stringency was ≥2 SNVs per read (Supplementary Fig. 8b and Supplementary Data 1, see Methods).

Requiring ≥2 SNVs per read, we turned to the datasets representing embryos ranging in age from the single, 2-, 4-, and 8-cell stage through the earliest days of embryogenesis and beyond. We observed no conclusive signal for *trans*-homolog contacts (Supplementary Figs. 9 and 10), consistent especially for the very earliest stages with previous indications that the maternal and paternal genomes are spatially segregated[25,26,48–50]; (Supplementary Fig. 9). More specifically, although the signal for *trans*-homolog contacts for some of the earlier stages seemed to hover above that expected from homolog misassignment (Supplementary Fig. 9) and thus raise the possibility of homolog pairing, this discordance between observed and expected *trans*-homolog contacts could also reflect artifacts arising from a non-uniform genomic density of SNVs and/or Hi-C biases (see[37]); we were unable to resolve this issue due to the sparsity of data (Supplementary Data 1). The data also raise the possibility of a higher order clustering of homologous as well as heterologous subtelomeric regions (Supplementary Fig. 10), reminiscent of the

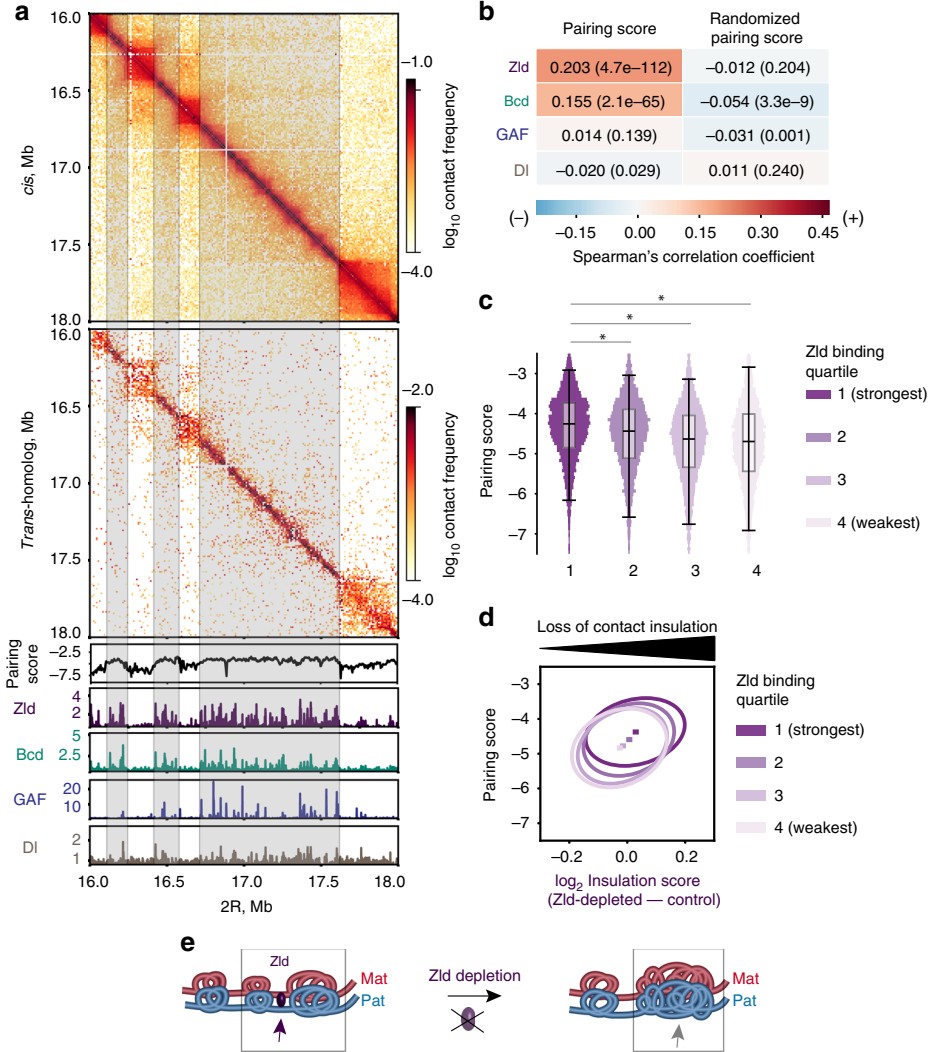

**Fig. 4** Homolog pairing is related to Zld-mediated opening of chromatin. **a** *Cis* (upper panel) and *trans*-homolog (second panel from the top) contact maps of a 2 Mb region (2 R:16-18 Mb). Lower panels, pairing score (PS) calculated using a 28 kb window at 4 kb resolution (black), ChIP-seq profiles of Zld (dark purple)[41], Bcd (green)[40], GAF (blue)[43], and DI (brown)[41]. Gray boxes, regions associated with elevated PS values in boundaries. **b** Correlation analyses between the PS values and the binding profiles of Zld, Bcd, GAF, and DI as determined using pairwise Spearman correlation. Control, randomized 200 kb chunks of the PS. Spearman correlation coefficients indicated in each box and by a heatmap; *P*-values in parentheses. **c** 25% of the strongest Zld binding coincides significantly with high PS regions, compared to the remaining Zld binding (*$P < 10^{-10}$, Mood's median test). **d** PS values *vs* loss of contact insulation upon Zld depletion, grouped by Zld binding strength quartile. Ovals represent the contours of 2D-Gaussian approximations, at a distance of 2 standard deviations from the mean (the dot). Regions with stronger Zld binding show both high PS values and increased loss of contact insulation upon Zld depletion. **e** Zld depletion affects insulation of domain boundaries at regions otherwise associated with high pairing. Thus, Zld may affect homologs to be further away from each other when insulation is lost

clustering of pericentromeric and subtelomeric regions seen previously[48]. While, again, the sparsity of data precluded our ability to determine such clustering reflected any degree of homolog pairing, it is worth noting that telomere as well as centromere clustering has been reported in embryos across several species[34,48,51] and may constitute a mechanism for initiating pairing[13–15,52]. In brief, although we found no definitive evidence of homolog pairing in early mouse embryos, our data nevertheless leave this issue open for future studies. Also, as repetitive sequences were excluded from our analyses, the potential of *trans*-homolog contacts at repeats remains to be explored.

## Discussion

To conclude, our study addressed the fundamental nature of chromosome pairing and, to this end, developed a robust method,

Ohm, for conducting haplotype-resolved Hi-C studies wherein interactions between homologous chromosomes are carefully vetted for homolog misassignment. This approach ensured the highest quality assignments of parental origin to Hi-C reads when distinguishing *cis* from *trans* contacts, and can be extended to any number of other methods[53].

Using Ohm, we obtained, for the first time, a genome-wide high-resolution view of homolog pairing during early *Drosophila* development. Our data revealed genome-wide juxtaposition of homologs along their entire lengths as well as a multi-layered organization of *trans*-homolog domains, domain boundaries, compartments, and interaction peaks. We also observed concordance of *trans*-homolog and *cis*- features, arguing that *cis* and *trans* interactions are structurally coordinated (see also[31]). This striking correspondence of *trans*-homolog and *cis* domain boundaries and interaction peaks may suggest similar

mechanisms of their formation. For instance, SMC complexes implicated in loop extrusion[54], formation of *cis* domains, and cohesion of sister chromatids in mammals may play a role in pairing and formation of *trans*-homolog domains in *Drosophila*. We pursue this line of thought in our companion paper[31].

Moreover, by layering *trans*-homolog contacts on models of genome organization that have relied primarily, if not solely, on *cis* contacts, our findings highlight how haplotype-resolved Hi-C can shed light on paradigms of genome organization and function that would otherwise remain hidden. For example, pairing may underlie some of the structural and regulatory differences between the 3D architecture of the *Drosophila* genome and that of organisms lacking pairing; indeed, the close proximity of homologs may influence the manner in which loops and domains are formed in *Drosophila* in addition to playing a role in gene regulation[2,31,55].

Our observations have further revealed a correlation between pairing and Zelda-mediated chromatin accessibility during zygotic genome activation. In this way, they champion a relationship between *trans*-homolog genome organization and key developmental decisions, aligning well with the growing recognition that pairing can play a potent role in gene regulation, even at some loci in mammals (reviewed by[1,2]). As such, it may be of significance that homologs are unpaired during the earliest moments of *Drosophila* embryogenesis[56] and then initiate pairing at a time coinciding with activation of the zygotic genome. This in mind, we examined data pertaining to early mouse embryo; as in *Drosophila*, murine maternal and paternal genomes are segregated in the earliest cell cycles[25,26,48–50], but little is known about the genome-wide status of pairing thereafter. Our analyses were further fueled by studies suggesting that pairing may be a general mechanism by which organisms detect structural heterozygosity in their genomes early in development, culling those that are deleteriously rearranged[57]. Although we observed some intriguing signals, ultimately, the sparsity of data did not allow us to draw strong conclusions regarding pairing in the mouse embryos, pointing to a need for additional haplotype-resolved Hi-C analyses and/or the resolution provided by imaging.

Our studies also suggest how pairing in *Drosophila* could shape early nuclear organization of the zygotic genome. For instance, as pairing brings similar loci together, it may enhance the compartmentalization of heterochromatic and euchromatic structures during phase separation in early *Drosophila* embryos[58]. Alternatively, the segregation of heterochromatin from euchromatin could facilitate homology searches during the initiation of pairing. Indeed, how homologs identify each other remains one of the most intriguing questions. For instance, do homologs become structurally similar because of pairing, or do they have comparable conformations prior to pairing, their similarities contributing to homolog recognition? As our results do not reveal striking differences between the *cis* maps of homologs, they are consistent with the latter scenario. However, because our *cis* maps likely represent a mix of paired and unpaired homologs, potential differences between the *cis* maps of homologs may only become apparent in younger embryos prior to pairing.

Finally, we note that the sensitivity of Ohm has potential usefulness also for the analysis of *trans* contacts between heterologous chromosomes, which would fall in the plane of $x = 0$ in our framework of precision (x), proximity (y), and continuity (z). Intriguingly, it may be that, in some instances, only one of two alleles participates in heterologous interactions (reviewed by[59]), and thus, here, our haplotype-resolved approach could be applied to dissect the parental origin of the interacting alleles. Of course, when our framework is extended to encompass the dimension of time, it should then be able to capture the dynamics of *trans*-homolog and *trans*-heterolog contacts through cell division, development, aging, and disease.

## Methods

**Selection of parental *Drosophila* lines.** To be able to distinguish homologs in sequencing data, we selected two parental lines with a high number of SNVs from *Drosophila melanogaster* Genetic Reference Panel (DGRP), which contains *Drosophila* lines inbred over 20 generations (see Supplementary Note 1).

**Genomic DNA isolation and library preparation.** 20 adult flies from each parental line (the Bloomington stock numbers 29652 and 29658) were homogenized with motor (Kimble-Kontes Pellet Pestle Cordeless Motor) and pestle (Kimble-Kontes) on ice. Genomic DNA was isolated using DNeasy Blood & Tissue Kit (Qiagen) according to manufacturers' recommendations. The integrity of the isolated genomic DNA (i.e. no degradation present) was verified by running a 1% agarose gel. Genomic DNA libraries were generated using Illumina TruSeq Nano DNA Library Preparation kit, and were 150 bp paired-end sequenced at the TUCF Genomics Facility using Illumina HiSeq2500.

**Collection and fixation of the F1 hybrid embryos.** The *Drosophila* handling was in accordance with the Harvard Medical School guidelines. The inbred DGRP-057 (maternal) and 439 (paternal) lines were mated to obtain the F1 hybrid embryos. To ensure that accurate parental genotypes were crossed, around 28,000 males and 28,000 virgin females were manually sorted. The F1 hybrid embryos were collected and fixed after three pre-lays followed by aging for 2–4 h time-point at 25 °C. To validate that the embryos were of the correct stage, and did not contain contaminants from older embryos, a small aliquot (~100 embryos) per collection was set aside, devitellinised, and stored in methanol at −20 °C to verify developmental stages of collection under microscope. If even a single older embryo was noted, collection was discarded, as that embryo may contribute more nuclei than 2–4 h embryo, depending on exact developmental difference between them. The remaining embryos from collections were snap-frozen in liquid nitrogen and stored at −80 °C.

**In situ Hi-C on *Drosophila* F1 hybrid embryos.** Hi-C protocol from Rao et al.[19] was adapted for the *Drosophila* embryos. 30 mg of snap-frozen fixed embryos were homogenized with pestle (Kimble-Kontes) in 500 μl of ice-cold lysis buffer, and incubated 30 min on ice. Upon cell lysis, the chromatin was digested using 500 U of DpnII restriction enzyme overnight at 37 °C. The DpnII overhangs were filled in with biotin mix during 1.5 h at 37 °C, and the chromatin was ligated for 4 h at 18 °C. RNA was degraded by adding 5 μl of 10 mg/ml RNase A (ThermoFisher Scientific) followed by incubation for 30 min at 37 °C. After degradation of proteins and crosslink reversal overnight at 68 °C, DNA was ethanol precipitated. DNA shearing was performed to obtain around 400–600 bp fragments with Qsonica Instrument (Q800R) using the following parameters: 30 s on/off, 15x cycles, 70% ampl. After size-selection to around 500 bp, biotin pull-down was performed with a minor modification. Namely, the size-selected DNA was eluted in 200 μl of 10 mM Tris-HCl pH 8.0, and was mixed with equal volume of 2x Binding Buffer (10 mM Tris-HCl pH 7.5, 1 mM EDTA, 2 M NaCl) containing pre-washed Dynabeads MyOne Streptavidin T1 beads (Life Technologies).

The library preparation[19] was performed with the following modifications. The adapter ligation was in 46 μl of 1x Quick ligation reaction buffer (NEB), 2 μl of DNA Quick ligase (NEB), and 2 μl of NEXTflex DNA Barcode Adapter (25 μM, NEXTflex DNA Barcodes −6, Bioo Scientific #514101). PCR reaction contained 23 μl of adaptor ligated DNA, 2 μl of NEXTflex Primer Mix (12.5 μM), and 25 μl NEBNext Q5 Hot Start HiFi PCR Master Mix (NEB), and was run using the program: 98 °C, 30 s, (98 °C, 10 s, 65 °C, 75 s) repeated 7 times, 65 °C, 5 min, hold at 4 °C. Purification of PCR products was performed by incubation for 15 min instead of 5 min after each addition of Agencourt AMPure beads. The final libraries were eluted in 15 μl of 10 mM Tris-HCl pH 8.0 after 15 min incubation at room temperature, and 15 min incubation on a magnet. The library quality was assessed using the High Sensitivity DNA assay on a 2100 Bioanalyzer system (Agilent Technologies). The libraries corresponding to two independent biological replicates were 150 bp paired-end sequenced at the TUCF Genomics Facility using Illumina HiSeq2500.

**FISH probes.** Oligo probes for heterochromatin repeats at chr2 (AACAC), chr3 (dodeca) and chrX (359)[60] were synthesized by Integrated DNA Technologies (IDT) with the following sequences and a fluorescent dye: AACAC (Cy3-AACA CAACACAACACAACACAACACAACACAACAC), dodeca (FAM488-ACGGG ACCAGTACGG), and 359 (Cy5-GGGATCGTTAGCACTGGTAATTAGCTGC).

The libraries for euchromatin probes 89D-89E/BX-C (315 kb), 2R16 (904 kb), 3R7 (844 kb), 3R19 (700 kb), 69 C (674 kb), and 28B (680 kb) were designed using Oligopaint technology (Supplementary Table 4)[17,61,62], and were amplified using T7 amplification (see section Oligopaint probe synthesis) with forward primers containing a site for secondary oligo annealing, and reverse primers containing a T7 promoter sequence (sequences are provided in Supplementary Table 5). DNA secondary oligos[61], which were used together with primary probes, had both 5′ and 3′ conjugated fluorophores (Supplementary Table 5).

**Oligopaint probe synthesis.** Oligopaints probes were synthesized using T7 amplification followed by reverse transcription[63]. Oligopaints libraries were first

amplified with Kapa Taq enzyme (Kapa Biosystems, 5 U/μl), and corresponding forward and reverse primers (without the site for secondary oligo annealing and the T7 promoter, i.e. without the underlined sequences from Supplementary Table 5) using linear PCR program as follows: 95 °C, 5 min, (95 °C, 30 s, 58 °C, 30 s, 72 °C, 15 s) repeated 25 times, 72 °C 5 min, hold at 4 °C. Upon clean up with DNA Clean & Concentrator-5 (DCC-5) kit (Zymo Research), this PCR was followed by another bulk up PCR using the same program, but this time to introduce the site for secondary oligo annealing to the forward strand, and the T7 promoter to the reverse strand (primers used with the underlined sequences from Supplementary Table 5). The PCR product with added T7 promoter was cleaned up using DNA Clean & Concentrator-5 (DCC-5) kit (Zymo Research), and became a template in T7 reaction to produce excess RNA with HiScribe T7 High Yield RNA Synthesis Kit (NEB) and RNAseOUT (ThermoFisher Scientific) overnight at 37 °C. RNA was then reverse transcribed into DNA with Maxima H Minus RT Transcriptase (ThermoFisher Scientific) in a 2 h reaction at 50 °C, followed by the inactivation of the RT enzyme at 85 °C for 5 min. Subsequently, RNA was degraded via alkaline hydrolysis (0.5 M EDTA and 1 M NaOH in 1:1) for 10 min at 95 °C, and ssDNA was purified with DNA Clean & Concentrator-5 (DCC-5) kit (Zymo Research), where DNA binding buffer was replaced with Oligo binding buffer (Zymo).

**FISH in embryos.** FISH in *Drosophila* embryos[17,61] was performed with modifications. Embryos were dechorionated with 50% bleach for 2.5 min, washed with 0.1% Triton X-100 in PBS, and fixed for 30 min in 500 μl of fix (PBS containing 4% formaldehyde, 0.5% Nonidet P-40, and 50 mM EGTA), plus 500 μl of heptane. Upon replacing the aqueous phase with methanol, embryos were vigorously shaken for 2 min. Finally, embryos were triple washed with 100% methanol, and stored at −20 °C in methanol.

Prior to FISH, embryos were gradually rehydrated in 2x SSCT (0.3 M NaCl, 0.03 M NaCitrate, 0.1% Tween-20). Subsequently, embryos were incubated for 10 min in 2x SSCT/20% formamide, followed by another 10 min in 2x SSCT/50% formamide, and then hybridized in a hybridization solution (2x SSCT, 50% formamide, 10% dextran sulphate, RNase) with primary Oligopaint probe set (200 pmol for heterochromatin and 300 pmol for euchromatin probes) for 30 min at 80 °C, and then at 37 °C overnight. After primary probe hybridization, two 30 min washes in 2x SSCT/50% formamide were performed at 37 °C. In a case secondary probe was used, secondary hybridization was performed between those two washes for 30 min in 2x SSCT/50% formamide with 300 pmol of secondary probe at 37 °C. The washes were continued in 2x SSCT/20% formamide for 10 min at RT, and with three rinses in 2x SSCT. The embryos were stained with Hoechst 33342 (Invitrogen), washed in 2x SSCT for 10 min at RT, quickly rinsed in 2X SSC, and mounted in SlowFade Gold antifade mountant (Invitrogen). Images were taken using a Zeiss LSM780 laser scanning confocal microscope with a ×63 oil NA 1.40 lens at 1024 × 1024 resolution.

**Analysis of FISH signals.** The analysis was performed by manually examining each section of Z-stack with the Zeiss ZEN Software. Distance in 3D space between two FISH signals was measured using the Ortho-distance function in the Zeiss ZEN Software. Two homologs were considered paired if 3D distance between centers of two FISH signals was ≤0.8 μm or there was only one FISH signal.

**Generation of Zld-depleted embryos.** The *UAS-shRNA-zld*[41] virgin females were crossed to the maternal triple driver Gal4 (*MTD-Gal4*, the Bloomington stock number 31777) males. The F1 heterozygous females *UAS-shRNA-zld /MTD-Gal4* were then mated to their sibling males, and the F2 Zld-depleted embryos were collected, and fixed as described in the section FISH in embryos. The control embryos were obtained using the same crossing scheme just by replacing the *MTD-Gal4* with the wild-type Canton S males. To validate that embryos were of the correct genotype, a small aliquot (~500 embryos) was inspected under microscope for both Zld-depleted and control samples.

**The construction of F1 diploid *Drosophila* genome.** We sequenced the two inbred parental *Drosophila* lines (DGRP-057 and DGRP-439) together with the hybrid PnM cell line[31] at the average coverage of 118, 117, and 396 reads per base pair, respectively[64]. We then detected the sequence variation of these three libraries using *bcftools*. In summary, we obtained high-quality normalized sequence variants as following:

1. Trimmed low-quality sequences with seqtk trimfq, aligned whole-genome paired-end reads against the reference dm3 genome using BWA mem, and removed aligned PCR duplicates with samtools −rmdup.
2. Piled alignments up along the reference genome with bcftools pileup −min-MQ 20 −min-BQ 20.
3. Called raw sequence variants from the pileups with bcftools call.
4. Normalized raw sequence variants with bcftools norm.
5. Selected only high-coverage high-quality normalized sequence variants using bcftools filter INFO/DP > 80 & QUAL > 200 & (TYPE = "SNV"|IDV > 1).

We then phased heterozygous PnM variants using bcftools isec. We picked high-confidence variants on the maternal autosomes by selecting heterozygous

PnM variants that were present among maternal DGRP-057 variants and absent among paternal DGRP-439 sequence variants (both homo- and heterozygous); the high-confidence paternal variants phasing was selected in an opposite manner.

Since PnM is a male line, for the maternal copy of chrX in F1 embryos, we considered only homozygous high-quality variants detected in the PnM cell line. To reconstruct the consensus sequence of the paternal copy of chrX in F1 embryos, we kept only homozygous variants detected in the paternal DGRP-439 *Drosophila* line.

Finally, we reconstructed the sequence of the F1 embryos with samtools consensus, using (a) the homozygous autosomal PnM SNVs, (b) the heterozygous phased autosomal PnM SNVs, and (c) the homozygous maternal chrX PnM SNVs as well as the homozygous SNVs detected in the paternal DGRP-439 *Drosophila* line.

**Mapping and parsing.** We started with sequences of Hi-C molecules and trimmed low-quality base pairs at both ends of each side using the standard mode of *seqtk trimfq* v.1.2-r94 (https://github.com/lh3/seqtk). Then we mapped the trimmed sequences to the reference dm3 genome or the constructed dm3-based F1 diploid genome using *bwa mem* v.0.7.15[65] with flags -SP.

We then extracted the coordinates of Hi-C contacts using the *pairtools parse* command line tool (https://github.com/mirnylab/pairtools). We only kept read pairs that mapped uniquely to one of the two homologous chromosomes and removed PCR duplicates using the standard mode of the *pairtools dedup* command line tool.

**Contact frequency (P(s)) curves.** We used unique Hi-C pairs to calculate the functions of contact frequency P(s) *vs* genomic separations. We grouped genomic distances between 10 bp and 10 Mb into ranges of exponentially increasing widths, with 8 ranges per order of magnitude. For every range of separations, we found the number of observed *cis*- or *trans*-homolog interactions within this range of separations and divided it by the total number of all loci pairs separated by such distances.

We used a similar technique to quantify the degree of coalignment of chromosomal arms due to Rabl configuration (Fig. 3h). Specifically, the Rabl configuration increased the contact frequency between pairs of loci located at the same relative positions along different chromosomal arms. We characterized this tendency by calculating a P(s) on *trans*-heterolog maps, where we calculated the position mismatch as $s = x_1 - x_2/L_2 \times L_1$, where $x_1$ and $x_2$ were the distances from each locus to the centromere and $L_1$ and $L_2$ were the lengths of the two chromosomal arms.

**Estimating the rate of Hi-C byproducts.** Each Hi-C dataset contains non-informative byproducts, i.e. sequenced DNA molecules that originate from a single DNA fragment, and thus do not carry any information on chromatin conformation (unlike informative Hi-C molecules that form via a ligation of two spatially proximal DNA fragments and capture a spatial contact) (Supplementary Fig. 3a, b). The two well-known types of Hi-C byproducts are: (i) unligated DNA fragments (termed "dangling ends") and (ii) self-ligated DNA fragments ("self-circles")[37].

"Dangling ends" are small pieces of intact unligated DNA that passed through all selection steps of the Hi-C protocol and ended up in the library of sequenced molecules (Supplementary Fig. 3b). "Dangling ends" appear as *cis* read pairs at s ~100–1000 bp (such molecules are left in the library after the size selection step of the Hi-C protocol), with the direction of the two reads of a read pair pointing towards each other along the reference genome.

"Self-circles" are pieces of DNA whose ends were ligated to each other and the resulting circular DNA had a break in another location, producing a linear piece of DNA (Supplementary Fig. 3b). "Self-circles" also appear in *cis*, but at longer separations, up to a few kb (this distance depends on the frequency of the restrictions sites along the genome) and the resulting paired read orientations point away from each other along the reference genome[37].

Because both of these types of byproducts were characterized by their narrow range of genomic separations and a specific mutual orientation of the paired end reads (Supplementary Fig. 3b), their abundance was estimated using P(s) curves for four possible combinations of the directionality of paired read ends (Supplementary Figs 3c and 4a). In such plots, "dangling ends" produced a characteristic $100 \times -1000x$ enrichment of inward read pairs below 1 kb separation and "self-circles" produced a weaker enrichment of outward read pairs at separations up to 10 kb, though their amount and typical genomic separation seemed to vary greatly between different experimental protocols (Supplementary Figs. 3c and 4a)[37].

During the work on this manuscript, we also discovered a third, previously undescribed type of a Hi-C byproduct. We found that the read pairs that aligned to the same genome strand (i.e. same-strand read pairs) were enriched at s < 500 bp. Upon closer examination, we found that such short-distance same-strand read pairs shared a similar unique pattern: while one of the two reads were typically fully aligned to some locus on the genome, the other read would split into two fractions, which would align to different locations. The inner, 3′ fraction mapped in *cis*, downstream and opposite to the read on the first side (as in a "dangling end");

while the outer, 5′ fraction overlapped the inner fraction, but went into the opposite direction (Supplementary Fig. 3d). This non-trivial alignment pattern suggested a scenario, where, during some stage of the Hi-C protocol, a DNA molecule formed a hairpin and was extended using itself as a template (Supplementary Fig. 3e). During the work on this manuscript, this type of a Hi-C byproduct was independently reported and thoroughly characterized in[66], who dubbed it as "hairpin loops". Another, older study reported formation of similar hairpin loops in whole-genome sequencing of ancient DNA[67].

The mechanism by which such "hairpin loops" form remains to be firmly established. Golloshi et al.[66] proposed that these "hairpin loops" were formed during the end repair stage of the Hi-C protocol via self-annealing of a pre-existing microhomology region followed by an elongation by the T4 polymerase.

If this hypothesis was true, then "hairpin loops" should form with equal probability on all Hi-C molecules, regardless of the orientation and relative location of the ligated fragments, as well as on "dangling ends" and "self-circles". The enrichment in short distance same-strand read pairs would then be most likely due to formation of "hairpin loops" on "dangling ends", which is a highly enriched substrate. Importantly, we can still identify the ligated fragments even in Hi-C molecules with "hairpin loops". First, the extra base pairs introduced by a "hairpin loop" map to the same location as the substrate (albeit at the opposite direction) and their alignment can serve as a proxy for the location of the substrate. Second, provided a sufficient long read length, we can simply ignore these extra pairs and instead align the inner fraction of the Hi-C molecule, which served as a substrate for the "hairpin loop". Thus, "hairpin loops" represent the most benign type of a Hi-C byproduct and their presence does not significantly reduce the number and identity of contacts extracted from a Hi-C library.

We produced an upper estimate of the amount of byproduct types in our sample, assuming that the true P(s) followed that of the same-strand read pairs at s > 1 kb, and was constant at s<1 kb. Using this assumption, we found that "dangling ends" constituted 39.5% of mapped and phased reads in our Hi-C librarys, "self-circles" were 0.3%, and "hairpin loops" formed from "dangling ends" made up 2.9% of the library.

**Binning and balancing of Hi-C data**. We aggregated unique Hi-C pairs into genomic bins of 1 kb and larger, using the cooler package (https://github.com/mirnylab/cooler). Low signal bins were excluded prior to balancing using the MADmax filter: we removed all bins, whose coverage was 7 genome-wide median deviations below the median bin coverage. Additionally, we removed all cis- and trans-homolog contacts at separations below 3 kb, since these reads pairs were dominated by non-informative Hi-C artifacts, unligated DNA fragments and ligation sites formed via self-circularization of DNA fragments[68]. We then balanced the obtained contact matrices via iterative correction (IC), i.e. equalized the sum of contacts in every row/column to 1.0.

We calculate observed/expected contact frequency maps (often abbreviated as O/E CF) in cis and trans-homolog by dividing each diagonal of an IC contact map by its chromosome-wide average value over non-filtered genomic bins.

**The definition of homolog misassignment**. When we aligned a read to a diploid genome, we assigned to it three bits of information: which pair of homologous chromosomes this read originated from; the allele, i.e. which of the two homologs it originated from, and its location along the chromosome. Importantly, we could assign the allele (or, the homolog) to a read only if it overlapped some sequence that is unique to one homolog, i.e. if it overlapped a SNV; otherwise, it was impossible to tell which of the two homologs the read originated from.

Even in the Drosophila and mouse models with the highest density of SNVs, the sequences of the two homologous chromosomes were still highly similar and contained only one SNV per 100–200 bp. Because of that, the majority of read pairs (150 bp) lacked allele assignment on one or both sides; and the allele assignment for the rest of pairs was based on one or a few SNVs, and thus was prone to errors.

**Sources of homolog misassignment**. The errors in allele assignment (below, homolog misassignment, HM) occur for two reasons: (A) due to an error in the sequence of the read, or (B) due to an error in the sequence of the genome. Below we discussed how such errors lead to HM. Also, note that we only discussed the case where diploid genome sequences only contained SNVs and did not analyze the effect of insertions, deletions and duplications.

(A) The type and rate of sequencing errors varies highly among different sequencing platforms[69]. In the case of the most popular short-read platform, Illumina, sequencing errors typically lead to substitutions at the rate around 0.1% per base pair[69]. Such a substitution could occasionally lead to a HM, if it occurs at a SNV site and the erroneously called base pair by chance matches the other SNV allele.

(B) Some types of errors in the sequence of a diploid genome may lead to HM. The possible cases are:

(B1) an error at a site without a SNV (e.g. the true sequence is A in both alleles, which we wrote as A-A, but in the genome database it is T-T)—in this case, reads coming from either homolog would contain a mismatch with the database. Such error does not lead to HM.

(B2) a missing SNV (e.g. true A-T, but in the database it is A-A). Such an error leads to mapping imbalance, i.e. reads from the homolog with the correct sequence would get mapped, while the reads from a homolog with a sequence error will be dropped. A missing SNV does not lead to HM.

(B3) a false positive SNV (e.g. true A-A, database A-T). In this case, reads from both homologs will be assigned to the same allele (the one matching the true sequence). A false positive SNV error leads to both HM and mapping imbalance.

(B4) an erroneous SNV (e.g. true A-T, database A-C). In this case, reads from the homolog containing an error will not be mapped, which will produce mapping imbalance. An erroneous SNV does not lead to HM.

(B5) missing heterozygosity of an SNV allele. Samples derived from populations of organisms can contain heterozygous SNVs, i.e. different organisms within the population may contain different base pairs at the same location on the same homolog. The issue, however, is that genomic databases can store only one allele per homolog (below, referred to as a "known" allele), while the other alleles become ignored (referred to as "unknown" alleles) (note that this issue can be solved by using graph genomes, which can store multiple overlapping variants per position[70]). The effects of heterozygosity on mapping depends on the nature of "known" and "unknown" alleles on the two homologs:

(B5.1) non-overlapping "known" and "unknown" alleles, when all alleles of one homolog are different from the allele on the other homolog (e.g., true A/C-T, database A-T). This scenario leads to a partial mapping imbalance, because the reads containing the "unknown" allele do not get mapped. This does not lead to HM.

(B5.2) overlapping "known" and "unknown" alleles, when the "unknown" allele on one homolog is the same as the "known" allele on the other homolog (e.g. true A/T-T, database A-T). This scenario leads to HM, because all reads containing the "unknown" allele will be interpreted as containing the "known" allele from the other homolog. Also, this scenario leads to a mapping imbalance.

**Misassigned cis pairs can contaminate trans-homolog signal**. HM is particularly problematic for studying the trans-homolog contact patterns. If we would assign a wrong homolog to one side of a cis Hi-C pair, we would misinterpret this pair as a trans-homolog contact.

Given that trans-homolog contacts in our sample were much less frequent than cis ones, even a small probability of HM could lead to a significant contamination of the trans-homolog contact map. Note, that a cross-contamination between the cis maps of the two homologs was much less likely, since it required simultaneous HM on both sides of a Hi-C pair.

**Estimating homolog misassignment using Hi-C byproducts**. The observed trans-homolog read pairs thus was a mixture of "true" trans-homolog contacts and cis pairs with a misassigned homolog on one side. For P(s) curves, this could be expressed as:

$$P_{trans-homolog}(s)^{obs} = (1 - P_{HM}) * P_{trans-homolog}(s)^{true} + P_{HM} * P_{cis}(s), \quad (1)$$

where $P_{trans-homolog}(s)^{obs}$ was the P(s) curve for the observed trans-homolog contacts, $P_{trans-homolog}(s)^{true}$ was that for the true trans-homolog contacts (i.e. Hi-C trans-homolog pairs that represented a true trans-homolog ligation) and $P_{cis}(s)$ was P(s) for cis contacts, and $P_{HM}$ was the probability of HM. Here, the probability of HM was defined per paired read (i.e. the probability per side is a $P_{HM}/2$); we assumed that each Hi-C pair had the same chance to get an erroneously assigned homolog, regardless of its genomic separation or orientation, i.e. that HM probability was the same for all Hi-C pairs. We also accounted for HM in trans-homolog read pairs, which were falsely interpreted as cis ones.

Estimating the probability of HM in a mapped Hi-C library was a challenging task. A priori, for any given trans-homolog read pair, we cannot tell if it was produced by a true trans-homolog ligation, or it resulted from HM of a cis pair. However, we could reduce the relative amount of homolog-misassigned pairs by increasing the stringency of homolog assignment. For that purpose, we selected pairs where both sides overlapped at least two or three SNVs and where read alignments had no mismatches with respect to the reference genome. We reasoned that, for such pairs, HM was less likely, since it required simultaneous errors at multiple SNVs.

For pairs with 2 + SNVs on each side, the shape of $P_{trans-homolog}(s)$ changed at separations <1 kb: the ~10-fold enrichment of trans-homolog inward pairs at s < 1 kb that was present in unfiltered data (Supplementary Fig. 4a), was greatly reduced after increasing the stringency of homolog assignment (Fig. 2c and Supplementary Fig. 4b). We tried increasing the stringency of homolog assignment further by requiring 3 + SNVs on each side, but this did not change the shape of $P_{trans-homolog}(s)$ (Supplementary Fig. 4c). This observation suggested that the homolog-misassigned cis pairs did not significantly contribute to $P_{trans-homolog}(s)$ for pairs with 2 + SNVs. Conversely, decreasing the accuracy of homolog assignment by remapping our data to less accurate DGRP SNV annotations (see Supplementary Note 1), increased the abundance of observed short-distance inward trans-homolog pairs (Supplementary

Fig. 4d). We concluded that the enrichment of short-distance inward *trans*-homolog pairs was due to homolog-misassigned inward *cis* pairs.

Inward *trans*-homolog pairs at $s < 1$ kb were particularly sensitive to contamination by homolog-misassigned *cis* pairs, because at these separations and directionalities *cis* pairs were the most abundant ($>10^4$ more abundant that *trans*-homolog pairs of other directionalities) due to "dangling ends". Thus, even rare HM at a probability as low as $\sim 10^{-4}$ would generate enough false *trans*-homolog pairs to match the true signal, and thus change the shape of inward $P_{trans\text{-}homolog}(s)$ at $s < 1$ kb. Thus, the ratio of $P_{trans\text{-}homolog}(s)/P_{cis}(s)$ for inward pairs at short separations could be used as an upper estimate for the probability of HM:

$$P_{HM} \leq <P_{trans-homolog}(s)^{inward}/P_{cis}(s)^{inward}>_{300bp<s<600bp}, \quad (2)$$

where $<\dots>_{300bp<s<600bp}$ was the geometric mean around the range of separations between 300 bp and 600 bp, corresponding to the "dangling end" peak of $P_{cis}(s)$.

Using this approach, we estimated the HM probability in our data at $\sim 0.17\%$ (Fig. 2d). Importantly, this was an *upper* estimate, since not all of the *trans*-homolog inward pairs at $s < 1$ kb were homolog-misassigned *cis* pairs. Finally, given this probability, misassigned *cis* contacts (the second term in Eq. (1)) made up only 5% of detected *trans*-homolog contacts at $s \sim 1$ kb separations (Fig. 2e) and even less at larger separations.

**Analysis of published haplotype-resolved mouse Hi-C data.** We tested our Ohm approach to haplotype-resolved Hi-C using data from two published haplotype-resolved Hi-C studies on mice[25,26].

For mapping, we reconstructed two sets of diploid genomes (Black6 x DBA/2J and Black6 x PWK/PhJ) using SNVs from the Mouse Genome Project[71]. The genome of Black6 parental mouse line did not require reconstruction, since it served as the basis for the reference mm10/GRCm38 genome. We reconstructed consensus genomes with bcftools consensus, using only homozygous SNVs that passed the quality control. We then mapped the publicly available sequenced Hi-C libraries to the diploid genomes (Black6 x DBA/2J for[26] and Black6 x PWK/PhJ for[25]) and extracted the positions of Hi-C contacts using the same approach as for our *Drosophila* data. For contact maps, we only used Hi-C pairs overlapping 2 + SNVs on each side. We did not perform IC on the produced contacts maps due to their sparsity. Non-ICed raw contact maps have to be interpreted with a degree of caution, since they are affected by the variation of sequencing visibility of loci due to an uneven distribution of SNVs, GC content, amount of open chromatin, etc.

In a naive interpretation, a decay of *trans*-homolog contact frequency $P_{trans\text{-}homolog}(s)$ with distance s is indicative of homolog pairing, since it means that pairs of loci in homologous positions make more contacts than loci in non-homologous positions. However, such decay can also be observed in samples without pairing, but due to severe contamination of *trans*-homolog contacts by homolog-misassigned *cis* pairs. To interpret the curves of *trans*-homolog contact frequency $P_{trans\text{-}homolog}(s)$, we compared them to a negative control, no-pairing curves $P_{trans\text{-}homolog}(s)^{no\text{-}pairing}$, which described the expected amount of observed *trans*-homolog contacts in samples without pairing, but in presence of HM. $P_{trans\text{-}homolog}(s)^{no\text{-}pairing}$ could be derived from Eq.(1) assuming $P_{trans\text{-}homolog}(s)^{true} = const$:

$$P_{trans-homolog}(s)^{no-pairing} \sim avg.cross-parent\ trans-heterolog\ CF + P_{HM} * P_{cis}(s), \quad (3)$$

here, we could estimate $P_{HM}$ using Eq. (2).

Thus, in order to claim that a sample had homolog pairing, we had to observe significantly more contacts than that predicted by $P_{trans\text{-}homolog}(s)^{no\text{-}pairing}$, at least in some range of separations.

**Analysis of published Hi-C data from *Drosophila* embryos.** To compare our data with other studies on the structure of chromosomes in *Drosophila* embryos, we re-analysed the publicly available Hi-C dataset from[33] using the same methods as we used for our own data.

**Pairing score.** To characterize the degree of pairing between homologous loci across the whole genome, we introduced a genome-wide statistics track called *pairing score (PS)*. The PS of a genomic bin is $log_2$ of average *trans*-homolog IC contact frequency between all pairs of bins within a window of $\pm W$ bins. For each genomic bin i, its pairing score with window size W was defined as:

$$PS^W(i) = log_2<CF_{m,n}>, \quad (4)$$

averaged over bins *m* and *n* between i-W-th and i + W-th genomic bins on different homologs of the same chromosome.

Importantly, under this definition, the PS quantified only contacts between homologous loci and their close neighbors and did not quantify pairing between non-homologous loci on homologous chromosomes.

The choice of the window size W was guided by the balance between specificity and sensitivity. Using a bigger window increased sensitivity, accumulating contacts across more loci pairs, while smaller windows increased specificity, allowing to see smaller-scale variation of homolog pairing; empirically, we found that, for our contact maps binned at 4 kb resolution, using a $7 \times 7$ bin window ($W = 3$) provided the optimal balance between specificity and sensitivity.

A close examination of the PS track revealed two important features: (i) the *Drosophila* genome was divided into regions that demonstrate consistently high, relatively similar, values of PS, followed by extended regions where PS dipped into lower values, (ii) switching between high- and low-PS regions seemed to occur around insulating boundaries.

**Insulation scores.** For every bin of a contact map binned at 4 kb resolution, we calculated the insulation score as the total number of normalized and filtered contacts formed across that bin by pairs of bins located on the either side, up to 5 bins (20 kb) away for Hi-C mapped to reference dm3 maps, and up to 10 bins (40 kb) away for haplotype-resolved Hi-C maps (see Supplementary Note 1). We then normalized the score by its genome-wide median. To find insulating boundaries, we detected all local minima and maxima in the log₂-transformed and then characterized them by their prominence (Billauer E. peakdet: Peak detection using MATLAB, http://billauer.co.il/peakdet.html). The detected minima in the insulation score corresponded to a local depletion of contacts across the genomic bin, were then called as insulating boundaries. We found empirically that the distribution of log-prominence of boundaries had a bimodal shape, and we selected all boundaries in the high-prominence mode above a prominence cutoff of 0.1 for Hi-C mapped to the reference dm3 map, and a cutoff of 0.3 for haplotype-resolved Hi-C maps. We called the insulating boundaries in *trans*-homolog contact maps using the same technique, requiring a minimal prominence of 0.3. Finally, we removed boundaries that were adjacent to the genomic bins that were masked out during IC.

To estimate the similarity of insulating boundaries detected in the *cis* and *trans*-homolog contact maps, we calculated the number of overlapping boundaries. We allowed for a mismatch up to four genomic 4 kb bins (16 kb total) between overlapping boundaries to account for the drift caused by the stochasticity of contact maps. We reported average percentage of overlapping boundaries. For instance, the average fraction (73.9%) that overlapping boundaries between *cis* maternal and *cis* paternal boundaries occupied within *cis* maternal only (74.2%), and *cis* paternal only boundaries (73.7%). Similarly, for replicates, we calculated the average percentage of overlapping boundaries for *cis* maternal replicates (75.4%), and also for *cis* paternal replicates (71.1%), and then provided a mean of the two values (73.2%). In reference dm3 Hi-C contact maps, the percentage of overlapping boundaries between the two replicates reached 92.3%, suggesting that the lower reproducibility of haplotype-resolved boundaries is due to technical and not biological variation between the two replicates.

We estimated the significance of an overlap between two given boundary sets by comparing it to the overlap expected by chance alone, given the sizes of the two sets. We estimated the latter by calculating an overlap between two similarly-sized random subsets of genomic bins visible in our Hi-C maps. To estimate how significantly the observed overlap deviates from the random expectation, we repeated randomization 10 times and compared the random overlaps to the observed overlap using a t-test.

**ChIP-seq.** We mapped the publicly available raw ChIP-seq data following the same procedure as used by the ENCODE consortium[72], (https://github.com/ENCODE-DCC/chip-seq-pipeline). To calculate the number of overlapping ChIP-seq peaks, we re-sized all peaks to 1 kb around their centers and used pyBedTools to calculate peak intersections[73].

To generate the tracks of Zld binding, we used ChIP datasets GSM1596215 and GSM1596216 and input datasets GSM1596216 and GSM1596220 from study GSE65441[41]. For Dl, we used ChIP datasets GSM1596223 and GSM1596227 and input datasets GSM1596224 and GSM1596228 from the same study GSE65441[41]. For GAF, we used ChIP datasets GSM614652 and input datasets GSM614653 and GSM614654 from study GSE23537[43]. For wild-type Bcd, we used ChIP datasets GSM1332670 and GSM1332671 and input datasets GSM1332672 and GSM1332673 from GSE55256[40]. For Bcd in *zld* mutants, we used ChIP datasets GSM1332674 and GSM1332675 and input datasets GSM1332676 and GSM1332677 from GSE55256[40]. Finally, for RNA Pol II, we used ChIP datasets GSM1596231 and GSM1596235 and input datasets GSM1596232 and GSM1596236 from GSE65441[41].

**GRO-seq.** We mapped the publicly available GRO-seq datasets (GSM1020093 and GSM1020094 from GSE41611[47]) using STAR[74] following the same procedure as used by the ENCODE consortium, (https://github.com/ENCODE-DCC/long-rna-seq-pipeline/tree/master/dnanexus).

**Correlation analyses between the PS and ChIP or GRO-seq.** The correlation analyses[75] were performed by first dividing the genome into 10 kb bins. In each bin, the fraction of sequence occupied by the PS, the respective ChIP or GRO-seq signal was determined. Then, for each bin genome-wide correlations were calculated between the PS values and a ChIP or GRO-seq signal of interest. The strength of correlation was reported using Spearman correlation coefficients and corresponding P-values in two flavors, either for a pairwise comparison between two features, or as a part of analyses that examined whether the correlation between two features was affected by co-correlation with a third feature. The Spearman correlation coefficients were also displayed using a heatmap. Control regions consisted of the randomized PS or the binding profiles for Zld, Bcd, GAF, and Dl,

where each respective feature was chunked into blocks of 200 kb size, that were randomly moved around.

**Correlation between the PS and Zld binding**. To visualize the relationship between PS and Zld binding, we plotted the distribution of PS for genomic bins from each of the four quartiles of the genome-wide distribution of Zld ChIP-seq signal.

We then tested if the effects of Zld binding on chromatin conformation were correlated with the pairing status of loci. We re-analyzed the published Hi-C datasets on Zld depletion[33] with the same computational methods as above and obtained normalized Hi-C maps at 4 kb resolution for WT and Zld-depleted embryos at nuclear cycle 14. Then, we calculated the change of the 20 kb-window insulation score upon Zld depletion, $\Delta ins^{Zld}$, and compared it to PS. For bins from each quartile of Zld ChIP-seq binding, we fitted the resulting 2D-distribution of $\Delta ins^{Zld}$ vs PS with a bivariate Gaussian. We then plotted the resulting Gaussians as ellipses corresponding to ±2 standard deviations from the mean along each of the independent axes of the distribution (i.e. Mahalanobis distance of 2).

**Sets of genes**. We obtained target gene lists for AP and DV factors from MacArthur et al. (1% FDR dataset)[45] and Zeitlinger et al.[44]. We retrieved locations of housekeeping genes from the Flybase by selecting genes with at least moderate expression level (RPKM > 11) across all assessed stages and tissues[33,46].

**Reporting summary**. Further information on research design is available in the Nature Research Reporting Summary linked to this article.

## Data availability

All raw sequencing data and extracted Hi-C contacts have been deposited in the Gene Expression Omnibus (GEO) repository under accession number "GSE121255 [https://www.ncbi.nlm.nih.gov/geo/query/acc.cgi?acc=GSE121255]". The publically available ChIP-seq and GRO-seq datasets were obtained using accession numbers provided in the Methods section. All other relevant data is available upon request. The source data underlying Fig. 1e and Supplementary Figs. 2a and 7d are provided as a Source Data file.

## Code availability

We performed all custom data analyses in Jupyter Notebooks[76], using matplotlib[77], numpy[78], and pandas[79] packages. We automated data analyses in command line interface using GNU Parallel[80]. The software used in this study is available at https://github.com/mirnylab/.

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

## Acknowledgements

We are grateful to the Wu and Mirny laboratories, participants of the Annual Northeast Regional Chromosome Pairing Conferences, the Lieberman Aiden laboratory, 4DN DCIC, in particular Giancarlo Bonora from the Noble laboratory, Brian J. Beliveau, Guillaume J. Filion, Mirko Francesconi, Ben Lehner, M. Jordan Rowley, and the Cavalli laboratory, especially Frédéric Bantignies, for discussions, the Furlong laboratory for Oregon-R fly collection, the Rushlow laboratory for *UAS-shRNA-zld* line, the TUCF Genomics Sequencing Core Facility, the Microscopy Resources on North Quad (MicRoN), and the Bloomington *Drosophila* Stock Center for *Drosophila* stocks. We apologize to the authors whose work we could not cite due to constrains on referencing. This work was supported by an EMBO Long-Term Fellowship (ALTF 186-2014) to J.E., a William Randolph Hearst Award to R.B.M., and awards from NIH/NCI (Ruth L. Kirschstein NRSA, F32CA157188) to E.F.J., NIH/NHGRI (R01 HG003143) to J.D. (Howard Hughes Medical Institute investigator), NIH/NIGMS (R01 GM114190) to L.A.M., and NIH/NIGMS (RO1GM123289, DP1GM106412, R01HD091797) and HMS to C-t.W. J.D., and L.A.M. acknowledge support from the National Institutes of Health Common Fund 4D Nucleome Program (U54 DK107980).

## Author contributions

J.E., J.A.A., and C-t.W. designed the experiments. A.G. designed Hi-C computational analyses with input from J.E., J.A.A., L.A.M., and C-t.W. J.D. and B.R.L. provided input in experimental design and data analysis. Experimental data were generated by J.E. and J.A.A., with the help of S.C.N., E.F.J., T.N.S., M.A.H. for sorting males and virgin females, and G.N. for Oligopaints design. J.E. and R.B.M. selected the parental lines. J.E., J.A.A., A.G., B.R.L., G.F., N.A., M.I., R.B.M., and W.S. performed computational analyses of the Hi-C data. J.E. performed FISH data analysis and Zld depletion experiments. J.E., J.A.A., A.G., G.F., L.A.M., and C-t.W. interpreted the data. J.E., J.A.A., A.G., L.A.M., and C.-t.W. wrote the paper with input from the other authors.

## Additional information

**Competing interests:** The authors declare no competing interests.

