## [Peer Review File · Nature Communications]

Reviewers' comments:

Reviewer #1 (Remarks to the Author):

Erceg et al. developed a haplotype-resolved Hi-C method by using hybrid embryos of two divergent *Drosophila* species. This study provided an evidence that homolog pairing cooccurs in genome-wide during embryogenesis (Figure 2A). Based on the observation that the organization of domains and boundaries of trans-homolog pairing is similar to cis-organization (Figure 3I-K), authors propose a model in which homolog pairing occurs by adopting a structure that resembles cis-configuration (Figure 3N). This idea is also supported by the accompanying study by the authors that uses the hybrid PnM cultured cell line (manuscript by Abed et al.). In addition, authors found correlation between the pairing probability and the enrichment of the pioneering factor Zelda, implicating that chromatin accessibility is a key determinant of pairing efficiency in the *Drosophila* genome.

#1: While this study and accompanying study differ in the materials they used (embryos vs cultured cells), two studies address the essentially same question by using the same cross of *Drosophila* species (DGRP-057 x DGRP-439). Moreover, major conclusions drawn by the study are largely overlapping with the accompanying manuscript by the authors. Thus, I would suggest authors to wrap up two independent manuscripts into a single paper.

#2: Can the authors comment how the chromatin opening by Zelda contributes to homolog pairing? In the accompanying paper, RNAi experiment suggested that Slimb and Topo II facilitate homolog pairing in *Drosophila*. Is there any functional interplay among these factors during establishment and maintenance of trans-homolog interactions? In addition, authors suggested that Bicoid morphogen is enriched in the highly-paired regions (Figure 4A and B). Is this simply because anterior-posterior patterning genes are more prone to be located in the structured regions comparing to dorsal-ventral patterning genes? Or alternatively, do authors think that Bicoid by itself mediate homolog pairing through some mechanism? To discriminate these possibilities, authors might want to perform FISH assay in the absence of Bicoid as they did for Zelda (Figure S6D).

#3: Recent Hi-C study by Hug et al., 2017 Cell reported that not only Zelda, but also cluster of housekeeping genes and Pol II are highly enriched in the TAD boundaries in early *Drosophila* embryos. To clarify the similarity and the difference in the mechanisms underlying cis- and trans-interactions, I would suggest authors to further explore the role of these factors by looking at available datasets.

Reviewer #2 (Remarks to the Author):

I was asked to review this paper because I also reviewed their other paper submitted to Nature Communications. I think the overall message are pretty much the same. The only difference is that one paper is using cell line, while this paper use 2-4 hours embryo. The cell line paper also contains more novel biological insights than the current submission, because they also identified essential regulators for homolog pairing and performed knock-down experiments.

I do not have much comments/questions toward this paper, as the message is simple and clear. They did Hi-C in F1 embryo and observed the same thing as reported in the other paper by the same group of authors.

The only question I have is: what are the interactions between 2L and 2R? And there are lots of inter-chromosomal interactions? And they have pretty clear patterns. In the cell line paper, they didn't report such patterns (see attached screenshot). Left panel is from this paper and the right panel is

from the other submitted manuscript.

Reviewer #3 (Remarks to the Author):

Summary:

Here, Erceg and colleagues first develop a very robust haplotype-resolved Hi-C method and then apply this method to *Drosophila* embryos to study homolog pairing. Interestingly, structures found in trans are also found in cis. This reviewer feels that the haplotype-resolved Hi-C approach is very rigorous and robust, and, therefore, comments below focus on the biological insights gained from such an approach. On this basis, after revision, this reviewer feels this manuscript is appropriate for publication in *Nature Communications*.

Major comments:

The authors seem surprised that homolog pairing in *Drosophila* occurs genome-wide, along the entire lengths of chromosomes. Although homolog pairing has not been characterized before by Hi-C, *Drosophila* is a very classic system for studying this aspect of chromosome biology, and, although microscopy studies are not genome-wide in comparison to Hi-C, work from the Wu and Sedat labs have characterized pairing at many loci. Additionally, a classic example from *Drosophila* are polytene chromosomes that occur in many tissues and clearly show homolog and chromatid pairing along their entire lengths. It might be worthwhile considering this in regards to the genomics results presented here.

The authors point out in the main text that they see trans-homolog interaction peaks, but no such peaks are visible in Figs. 3I-K and only one peak is seen in Fig. S5A-C. Can the authors please expand upon trans-homolog peaks? There are certainly fewer peaks/loops in *Drosophila* than in mammals and *Drosophila* looping seems to be distinct from CTCF-dependent mammalian looping (Cubenas-Potts et al., 2017; Eagen et al., 2017). Is this related to why so few trans-homolog peaks are observed?

In Fig. 4A the Zelda and Bicoid ChIP-seq tracks look similar, so it is hard to determine which is more related to pairing. The correlation analysis in Fig. 4B suggests it is Zelda, but the Spearman correlation coefficient, though significant by P-value, is low. Besides using randomized chunks of PS as a control, it would help to randomize the Zld, Bcd, GAF, and DI binding profiles as an additional control.

On lines 328-330 of the conclusion the authors comment that "pairing may underlie some structural and regulatory differences between the 3D architecture of the *Drosophila* genome architecture and that of other organisms that lack pairing". What are these structural and regulatory differences?

Minor comments:

Like in mammalian systems, on diagonal boxes of enriched contact frequency have been termed TADs in *Drosophila*. Why are the authors not considering the domains they observe, particularly the cis domains, TADs?

The authors analyze data from Zelda depleted embryos to understand the relationship between homolog pairing and Zelda-mediated chromatin opening. In Zelda-depleted embryos there is a decrease in boundary strength at regions of high PS and strong Zelda binding. Since such boundaries

are also seen in cis are the Zelda-mediated changes more related to the cis domains, with the trans interactions a consequence or indirect effect of changes in cis?

The authors compare Fig. 3L to Fig. 3M. Fig. 3M is an observed/expected map, is Fig. 3L also an observed/expected map? If not, this comparison can't be made.

The manner in which the section on mammalian homolog pairing was presented left the impression that no conclusions could be drawn in regards to mammalian homolog pairing so it seems like this discussion does not add much to the current manuscript.

Perhaps the reviewer missed this, but are there trans-homolog compartments?

Reviewers' comments:

Reviewer #1 (Remarks to the Author):

Erceg et al. developed a haplotype-resolved Hi-C method by using hybrid embryos of two divergent *Drosophila* species. This study provided an evidence that homolog pairing cooccurs in genome-wide during embryogenesis (Figure 2A). Based on the observation that the organization of domains and boundaries of trans-homolog pairing is similar to cis-organization (Figure 3I-K), authors propose a model in which homolog pairing occurs by adopting a structure that resembles cis-configuration (Figure 3N). This idea is also supported by the accompanying study by the authors that uses the hybrid PnM cultured cell line (manuscript by Abed et al.). In addition, authors found correlation between the pairing probability and the enrichment of the pioneering factor Zelda, implicating that chromatin accessibility is a key determinant of pairing efficiency in the *Drosophila* genome.

#1: While this study and accompanying study differ in the materials they used (embryos vs cultured cells), two studies address the essentially same question by using the same cross of *Drosophila* species (DGRP-057 x DGRP-439). Moreover, major conclusions drawn by the study are largely overlapping with the accompanying manuscript by the authors. Thus, I would suggest authors to wrap up two independent manuscripts into a single paper.

*We thank the Reviewer for raising this point, as we now see that we did not do a good job of highlighting how the two manuscripts are different. In particular, while the biological messages may be related, the purpose and focus of the two papers are distinct in conceptual ways, and thus, we are hoping to keep the narratives separate. In particular, Erceg et al. aims to achieve a global perspective by establishing and implementing a technology that can be applied across species to provide haplotype-resolved Hi-C information regarding trans-homolog as well as trans-heterolog interactions, all with an emphasis on embryonic development. In contrast, AlHaj Abed et al. focuses on providing an analysis of the detailed structures of pairing, the relationship of those structures to gene expression, and potential mechanisms of pairing in *Drosophila*, with all its analyses being carried out in a cell line. We believe that, by separating these messages, each manuscript is stronger and more compelling as well as more approachable to audiences with diverse interests. Below, please find the more substantial changes we have made to more clearly highlight the distinctive goals of Erceg et al. (in blue and underlined).*

Page 2, line 31: We first addressed long-standing questions regarding the structure of embryonic homolog pairing and, to this end, developed a haplotype-resolved Hi-C approach to minimize homolog misassignment and thus robustly distinguish trans-homolog from cis contacts.

*Page 2, line 37: We also found a significant genome-wide correlation between pairing, transcription during zygotic genome activation, and binding of mediated by the pioneer factor Zelda. Our findings reveal a complex, highly structured organization underlying homolog pairing, first discovered a century ago in *Drosophila*. Finally, we demonstrated the versatility of our haplotype-resolved approach by applying it to mammalian embryos.*

Page 3, line 54: Although best known in meiotic cells (reviewed by^{2,3}), homolog pairing can also occur in somatic cells and influence gene expression via phenomena such as transvection (reviewed by^{2,4-8}). In *Drosophila*, somatic pairing increases extensively from early embryogenesis to adulthood, making this organism an ideal system for studying trans interactions (reviewed by^{2,4-8}). What, for example, is the structure of homolog pairing in early embryos?

Page 4, line 78: Here, we asked how pairing initiates in early *Drosophila* embryos, when maternal and paternal genomes first meet. Although the imaging of individual loci has documented the initiation of pairing during early embryogenesis^{20,24}, the global structure of pairing and its genome-wide relationship to fundamental developmental programs has remained elusive. To that end, we took advantage of single nucleotide variants (SNVs) to enable haplotype-resolved Hi-C. Haplotype-resolved Hi-C has previously been used to study cis interactions in mammalian systems²⁵⁻³⁶ as well as trans-homolog interactions in yeast^{37,38} and, here, we developed a method (Ohm) to much more confidently identify those trans interactions that occur between homologous chromosomes. Importantly, our approach allowed us to detect both short- and long-range interactions, that is, interactions spanning genomic distances as small as a few kilobases to several megabases. Excitingly, haplotype-resolved Hi-C as well as Ohm can be applied to any organism whose genome sequence has been haplotype-resolved, and we demonstrate their generality by applying them also to studies^{32,33} of early mammalian embryos.

Strikingly, our Ohm approach reveals that homolog pairing is extensive along entire lengths of chromosomes at short- as well as long-range distances and is highly structured, with trans-homolog domains, domain boundaries, compartments, and interaction peaks. We reveal the coincidence between trans-homolog features and the positions and sizes of analogous cis features and then provide a three-axis framework of precision, proximity, and continuity to accommodate all types of inter-chromosomal interactions, including homolog pairing. These findings complement those of our companion study (AlHaj Abed, Erceg, Goloborodko et al. bioRxiv³⁹), which, by generating and then using a *Drosophila* hybrid cell line with a high degree of pairing, enabled a much more in-depth analysis of the structure of pairing. In particular, this other study observed at least two forms of pairing and correlations between these forms and the active or repressed state of chromatin. In our current study, we also explore the relationship between pairing and the developmental programs in play in the early embryo during the initial stages of pairing. We find that pairing correlates with the opening of chromatin mediated by the pioneer factor Zelda during zygotic genome activation.

Page 7, line 173: This approach, which we call Oversight of homolog misassignment (Ohm), allowed us to estimate the probability of homolog misassignment as only ~0.17% when requiring at least 1 SNV per read (Fig. 2d, see Supplementary methods).

Page 15, line 348: To conclude, our study addressed the fundamental nature of chromosome pairing and, to this end, developed a robust method, Ohm, for conducting haplotype-resolved Hi-C studies wherein interactions between homologous chromosomes are

carefully vetted for homolog misassignment. This approach ensured the highest quality assignments of parental origin to Hi-C reads when distinguishing cis from trans contacts. As such, Ohm can be extended to a variety of methods such as DNase Hi-C⁶⁷, Micro-C⁶⁸, Tri-C⁶⁹, as well as other variations of Hi-C⁷⁰ in order to examine the parental origin of contacts in ensemble and single-cell Hi-C⁷¹, taking into account even states of DNA methylation⁷².

Page 16, line 379: Our observations have further revealed a correlation between pairing and Zelda-mediated chromatin accessibility during zygotic genome activation. In this way, they champion a relationship between trans-homolog genome organization and key developmental decisions, aligning well with the growing recognition that pairing can play a potent role in gene regulation, even at some loci in mammals (reviewed by^{5,7}). As such, it may be of significance that homologs are unpaired during the earliest moments of Drosophila embryogenesis^{75,76} and then initiate pairing at a time coinciding with activation of the zygotic genome. This in mind, we examined data pertaining to early mouse embryo; as in Drosophila, murine maternal and paternal genomes are segregated in the earliest cell cycles^{32,33,62-64}, but little is known about the genome-wide status of pairing thereafter. Our analyses were further fueled by studies suggesting that pairing may be a general mechanism by which organisms detect structural heterozygosity in their genomes early in development, culling those that are deleteriously rearranged⁷⁷⁻⁸⁰ (bioRxiv⁸¹ and reviewed by⁷). Although we observed some intriguing signals, ultimately, the sparsity of data did not allow us to draw strong conclusions regarding pairing in the mouse embryos, pointing to a need for additional haplotype-resolved Hi-C analyses and/or the resolution provided by imaging.

Our studies also suggest how pairing in Drosophila could shape early nuclear organization of the zygotic genome. For instance, as pairing brings similar loci together, it may enhance the compartmentalization of heterochromatic and euchromatic structures during phase separation in early embryos⁸². Alternatively, the segregation of heterochromatin from euchromatin could facilitate homology searches during the initiation of pairing. Indeed, how homologs identify each other remains one of the most intriguing questions. For instance, do homologs become structurally similar because of pairing, or do they have comparable conformations prior to pairing, their similarities contributing to homolog recognition. As our results do not reveal striking differences between the cis maps of homologs, they are consistent with the latter scenario. However, because our cis maps likely represent a mix of paired and unpaired homologs, potential differences between the cis maps of homologs may only become apparent in younger embryos prior to pairing.

Finally, we note that the sensitivity of Ohm has potential usefulness also for the analysis of trans contacts between heterologous chromosomes; trans-heterologous contacts, such as those observed in T cells⁸³ or olfactory sensory neurons⁸⁴, would fall in the plane of $x = 0$ in our framework of precision (x), proximity (y), and continuity (z). Intriguingly, it may be that, in some instances, only one of two alleles participates in heterologous interactions (⁸⁵ and reviewed by⁸⁶), and thus, here, our haplotype-resolved approach could be applied to dissect the parental origin of the interacting alleles. Of course, when our framework is extended to encompass the dimension of time, it should then be able to capture the dynamics of trans-homolog and trans-heterolog contacts through cell division, development, aging, and disease.

#2: Can the authors comment how the chromatin opening by Zelda contributes to homolog pairing? In the accompanying paper, RNAi experiment suggested that Slmb and Topo II facilitate homolog pairing in Drosophila. Is there any functional interplay among these factors during establishment and maintenance of trans-homolog interactions?

Thank you for raising this very interesting point. Although disruption of Zld, Slmb, or TopoII can in each case disrupt pairing, too little is known about the mechanistic underpinnings of pairing to know whether there is interplay between these factors or whether they act independently. However, encouraged by the Reviewer's question, we have now added additional speculation about these factors.

Page 12, line 296: Thus, Zld-mediated chromatin accessibility and transcription appear correlated with the establishment of homolog pairing in the early embryo (Fig. 4e), reinforcing the notion that the establishment of homolog pairing may bear a relationship to genome function during key developmental events. For example, Zld-mediated opening of genomic regions may facilitate homologous sequences finding each other when pairing is initiated in early embryogenesis. Whether Zelda plays a direct or indirect role in initiating pairing, however, will require additional study, as will the interplay between Zld and other factors involved in pairing and other forms of chromosome organization.

In addition, authors suggested that Bicoid morphogen is enriched in the highly-paired regions (Figure 4A and B). Is this simply because anterior-posterior patterning genes are more prone to be located in the structured regions comparing to dorsal-ventral patterning genes? Or alternatively, do authors think that Bicoid by itself mediate homolog pairing through some mechanism? To discriminate these possibilities, authors might want to perform FISH assay in the absence of Bicoid as they did for Zelda (Figure S6D).

We appreciate the Reviewer's suggestions regarding the relationship between pairing and developmental events. We explored these questions, and our results are included in Figure S6 and as new text. Briefly, we found that Zld may contribute to an indirect link between Bcd, itself, and pairing, and thus, we did not proceed with assessing pairing by FISH in the absence of Bcd. Our findings are described in the new text as shown in the excerpts, below.

Page 11, line 261: We correlated the binding of these factors, as revealed by ChIP-seq datasets^{52,54,57}, to local variations in the degree of pairing as characterized by the 'pairing score (PS)', defined as the log₂ trans-homolog contact frequency within a 28 kb window along the diagonal of Hi-C maps (Supplementary methods). Excitingly, regions associated with elevated PS values were significantly enriched for the binding of Zld and Bcd ($P < 10^{-10}$, Fig. 4a, b and Supplementary Fig. 6a). Using partial correlation analyses, we found that Zld contributes significantly to the correlation between the binding of Bcd and PS values, as well as between the binding of Dl and PS values (Supplementary Fig. 6b), consistent with a role of Zld in Bcd- and Dl-dependent binding and gene activation^{52,54}. Interestingly, we observed that genes^{58,59} associated with both AP and DV patterning are similarly associated with higher PS values (Supplementary Fig. S6c). Moreover, loss of Bcd binding in Zld-depleted embryos⁵² was significantly enriched at regions otherwise associated with high PS values ($P = 1.21 \times 10^{-68}$;

Supplementary Fig. S6d, e). However, in contrast to the binding of Bcd, neither the binding of GAF nor that of Dl were correlated with PS values ($P = 0.139$, $P=0.029$, respectively; Fig. 4b), though the correlation of Dl was affected by Zld (Supplementary Fig. 6b). Thus, our data suggest that Zld-mediated early opening of chromatin, including the binding of Bcd, is strongly correlated with pairing in embryos, unlike the binding of GAF or Dl.

Page 26, line 587 (in Supplementary Methods): The strength of correlation was reported using Spearman correlation coefficients and corresponding P-values in two flavors, either for a pairwise comparison between two features, or as a part of analyses that examined whether the correlation between two features was affected by co-correlation with a third feature. The Spearman correlation coefficients were also displayed using a heatmap.

#3: Recent Hi-C study by Hug et al., 2017 Cell reported that not only Zelda, but also cluster of housekeeping genes and Pol II are highly enriched in the TAD boundaries in early Drosophila embryos. To clarify the similarity and the difference in the mechanisms underlying cis- and trans-interactions, I would suggest authors to further explore the role of these factors by looking at available datasets.

We agree with the Reviewer and have now extended our analyses to include other available datasets. The outcome of these studies is presented in Figures S6c and S7e, and in new text.

Page 12, line 291: Third, we found that Zld depletion trended strongly toward reduced pairing, with the reduction being significant at 4 out of 6 loci assayed ($P < 2.56 \times 10^{-3}$, Supplementary Fig. 7c, d). Finally, we also observed strong correlations between PS values and features known to be enriched at domain boundaries, such as RNA Pol II⁴², housekeeping genes^{42,60}, and nascent zygotic gene products⁶¹ (Supplementary Fig. 6c, 7e). Thus, Zld-mediated chromatin accessibility and transcription appear correlated with the establishment of homolog pairing in the early embryo (Fig. 4e), reinforcing the notion that the establishment of homolog pairing may bear a relationship to genome function during key developmental events. For example, Zld-mediated opening of genomic regions may facilitate homologous sequences finding each other when pairing is initiated in early embryogenesis. Whether Zelda plays a direct or indirect role in initiating pairing, however, will require additional study, as will the interplay between Zld and other factors involved in pairing and other forms of chromosome organization.

Reviewer #2 (Remarks to the Author):

I was asked to review this paper because I also reviewed their other paper submitted to Nature Communications. I think the overall message are pretty much the same. The only difference is that one paper is using cell line, while this paper use 2-4 hours embryo. The cell line paper also contains more novel biological insights than the current submission, because they also identified essential regulators for homolog pairing and performed knock-down experiments.

I do not have much comments/questions toward this paper, as the message is simple and clear. They did Hi-C in F1 embryo and observed the same thing as reported in the other paper by the same group of authors.

We are grateful to this Reviewer for taking the time to read both manuscripts and letting us know that the narrative of this one is clear. We also appreciate the feedback that the two manuscripts seemed very similar, as we had intended the two to convey different, although complementary, messages and see now that we did not achieve this goal. If we may, we would like to refer this Reviewer to our response on point #1 from the Reviewer #1, who had a similar comment.

1. The only question I have is: what are the interactions between 2L and 2R? And there are lots of inter-chromosomal interactions? And they have pretty clear patterns. In the cell line paper, they didn't report such patterns (see attached screenshot). Left panel is from this paper and the right panel is from the other submitted manuscript.

Yes, thank you, we should make an explicit connection between our comment about the Rab1 configuration and these interactions between 2L and 2R, thank you. We have now made the following modification to the text:

*Page 9, line 216: This framework can also be used to describe trans interactions between heterologous chromosomes, particularly prominent examples being those arising from the Rab1 polarization of centromeres and telomeres, appearing in some Hi-C maps, including ours, as contacts along non-homologous arms (e.g. 2L and 2R, as well as 3L and 3R)^{42,44,45,49}; the Rab1 configuration may facilitate homolog pairing by reducing search space within the nucleus (Fig. 3f,²⁰⁻²²). Note, however, that trans-homolog contacts were tighter and/or more frequent than trans-heterolog contacts (Fig. 3g, h), emphasizing the prevalence of homolog pairing in *Drosophila*.*

Reviewer #3 (Remarks to the Author):

Summary:

Here, Erceg and colleagues first develop a very robust haplotype-resolved Hi-C method and then apply this method to *Drosophila* embryos to study homolog pairing. Interestingly, structures found in trans are also found in cis. This reviewer feels that the haplotype-resolved Hi-C approach is very rigorous and robust, and, therefore, comments below focus on the biological insights gained from such an approach. On this basis, after revision, this reviewer feels this manuscript is appropriate for publication in Nature Communications.

Thank you very much for these positive comments. We are thrilled that the Reviewer has enjoyed our story.

Major comments:

1. The authors seem surprised that homolog pairing in *Drosophila* occurs genome-wide, along the entire lengths of chromosomes. Although homolog pairing has not been characterized before by Hi-C, *Drosophila* is a very classic system for studying this aspect of chromosome biology, and, although microscopy studies are not genome-wide in comparison to Hi-C, work from the Wu and Sedat labs have characterized pairing at many loci. Additionally, a classic example from *Drosophila* are polytene chromosomes that occur in many tissues and clearly show homolog and chromatid pairing along their entire lengths. It might be worthwhile considering this in regards to the genomics results presented here.

We agree with the Reviewer and see now how our narrative did not accurately represent our thoughts. What we had hoped to convey was that, until recently, most observations of homolog pairing in diploid cells were derived through the imaging of specific loci and, while polytene chromosomes show extensive pairing between homologs, this pairing of up to 2,000 homologs does not easily pertain to the situation in diploid nuclei. Furthermore, as there had been no genome-wide haplotype-resolved Hi-C characterization of homolog pairing when our first maps of *Drosophila* pairing were obtained, we were not prepared for the degree of pairing to be so extreme as to render the trans-homolog maps to be strikingly similar to the cis maps even in early embryos, where contacts between homologs are just initiating. Indeed, we have found audiences to whom we have presented our findings to be similarly surprised by the outcome of our Hi-C maps.*

**Note that, during the review and revision process of our paper, a publication examining very short-range allelic interactions was published¹⁹. Although this paper did not use SNVs and, thus, could not distinguish homolog pairing from interactions between sister chromatids, we were nevertheless intrigued by its analyses and have thus described it in our paper. Because of this, we have altered our narrative in terms of the “surprise” and, in fact, are pleased with the greater emphasis now on the structure of paired homologs when pairing is just initiating.*

*Page 3, line 54: Although best known in meiotic cells (reviewed by^{2,3}), homolog pairing can also occur in somatic cells and influence gene expression via phenomena such as transvection (reviewed by^{2,4-8}). In *Drosophila*, somatic pairing increases extensively from early embryogenesis to adulthood, making this organism an ideal system for studying trans*

interactions (reviewed by^{2,4,8}). What, for example, is the structure of homolog pairing in early embryos?

Page 3, line 70: These studies are consistent with long-standing discussions about the establishment, maintenance, and stability of pairing as well as the relationship between pairing and transcription (reviewed by^{2,4-8,17,18}). Nevertheless, questions regarding the extent and structure of pairing remain. More recently, Hi-C-derived ligation products with overlapping sequences have been used to infer short-range interactions between homologs or sister chromatids in a tetraploid Kc₁₆₇ cell line and thus a correlation between short-range interactions and active regions¹⁹.

Here, we asked how pairing initiates in early Drosophila embryos, when maternal and paternal genomes first meet. Although the imaging of individual loci has documented the initiation of pairing during early embryogenesis²⁰⁻²⁴, the global structure of pairing and its genome-wide relationship to fundamental developmental programs has remained elusive. To that end, we took advantage of single nucleotide variants (SNVs) to enable haplotype-resolved Hi-C. Haplotype-resolved Hi-C has previously been used to study cis interactions in mammalian systems²⁵⁻³⁶ as well as trans-homolog interactions in yeast^{37,38} and, here, we developed a method (Ohm) to much more confidently identify those trans interactions that occur between homologous chromosomes. Importantly, our approach allowed us to detect both short- and long-range interactions, that is, interactions spanning genomic distances as small as a few kilobases to several megabases. Excitingly, haplotype-resolved Hi-C as well as Ohm can be applied to any organism whose genome sequence has been haplotype-resolved, and we demonstrate their generality by applying them also to studies^{32,33} of early mammalian embryos.

Strikingly, our Ohm approach reveals that homolog pairing is extensive along entire lengths of chromosomes at short- as well as long-range distances and is highly structured, with trans-homolog domains, domain boundaries, compartments, and interaction peaks.

Page 8, line 188: In addition to displaying the expected central cis diagonal, it revealed prominent “trans-homolog” diagonals that extended across the entire length of each chromosome (Fig. 3a, black arrows). This observation demonstrated that interactions between homologs, including those within a few kilobases and up to several megabases, extend genome-wide in early embryos at the time when pairing is being initiated.

2. The authors point out in the main text that they see trans-homolog interaction peaks, but no such peaks are visible in Figs. 3I-K and only one peak is seen in Fig. S5A-C. Can the authors please expand upon trans-homolog peaks? There are certainly fewer peaks/loops in Drosophila than in mammals and Drosophila looping seems to be distinct from CTCF-dependent mammalian looping (Cubenas-Potts et al., 2017; Eagen et al., 2017). Is this related to why so few trans-homolog peaks are observed?

We appreciate the Reviewer pointing out the discrepancy between the text and the figures and have now added additional examples of trans-homolog interaction peaks (see the figure below this response). There are, indeed, fewer loops than one might expect, and one explanation

is that pairing is only just initiating at time at which we collected our embryos (2-4 hours). We do report ~40 loops in our haplotype-resolved maps, nonetheless.

Of note, we address trans-homolog loops more extensively in our companion study by AlHaj Abed et al. and, pending agreement by this Reviewer, would like to defer additional consideration of loops to this other manuscript. This would both emphasize the difference between the two studies as well as strengthen our discussion of loops. For convenience of the Reviewer, however, we will summarize here that AlHaj Abed et al. focused on the PnM hybrid cell line and reported ~70 loops that more often than not were detected in both trans-homolog and cis Hi-C maps. We believe that this higher incidence of trans-homolog loops in PnM cells may be related to the stronger, more established pairing in PnM cells, as compared to that of early embryos. AlHaj Abed et al. also discusses the potential differences between *Drosophila* and mammals in terms of loop formation.

Fig. Trans-homolog, cis maternal, and cis paternal maps for regions on 2L and 2R. Interactions peaks are encircled.

3. In Fig. 4A the Zelda and Bicoid ChIP-seq tracks look similar, so it is hard to determine which is more related to pairing. The correlation analysis in Fig. 4B suggests it is Zelda, but the

Spearman correlation coefficient, though significant by P-value, is low. Besides using randomized chunks of PS as a control, it would help to randomize the Zld, Bcd, GAF, and Dl binding profiles as an additional control.

We agree with the Reviewer and have now provided additional analyses as new panels a, b, and d in Figure S6. We have also modified the text accordingly.

Page 11, line 261: We correlated the binding of these factors, as revealed by ChIP-seq datasets^{52,54,57}, to local variations in the degree of pairing as characterized by the ‘pairing score (PS)’, defined as the log₂ trans-homolog contact frequency within a 28 kb window along the diagonal of Hi-C maps (Supplementary methods). Excitingly, regions associated with elevated PS values were significantly enriched for the binding of Zld and Bcd ($P < 10^{-10}$, Fig. 4a, b and Supplementary Fig. 6a). Using partial correlation analyses, we found that Zld contributes significantly to the correlation between the binding of Bcd and PS values, as well as between the binding of Dl and PS values (Supplementary Fig. 6b), consistent with a role of Zld in Bcd- and Dl-dependent binding and gene activation^{52,54}. Interestingly, we observed that genes^{58,59} associated with both AP and DV patterning are similarly associated with higher PS values (Supplementary Fig. S6c). Moreover, loss of Bcd binding in Zld-depleted embryos⁵² was significantly enriched at regions otherwise associated with high PS values ($P = 1.21 \times 10^{-68}$; Supplementary Fig. S6d, e). However, in contrast to the binding of Bcd, neither the binding of GAF nor that of Dl were correlated with PS values ($P = 0.139$, $P = 0.029$, respectively; Fig. 4b), though the correlation of Dl was affected by Zld (Supplementary Fig. 6b). Thus, our data suggest that Zld-mediated early opening of chromatin, including the binding of Bcd, is strongly correlated with pairing in embryos, unlike the binding of GAF or Dl.

4. On lines 328-330 of the conclusion the authors comment that “pairing may underlie some structural and regulatory differences between the 3D architecture of the Drosophila genome architecture and that of other organisms that lack pairing”. What are these structural and regulatory differences?

We appreciate the Reviewer’s encouragement to include more information that would be beneficial here. Thus, we have now added clarification and, as above, also refer the reader to AlHaj Abed et al.

Page 16, line 370: Moreover, by layering trans-homolog contacts on models of genome organization that have relied primarily, if not solely, on cis contacts, our findings highlight how haplotype-resolved Hi-C can shed light on paradigms of genome organization and function that would otherwise remain hidden. For example, pairing may underlie some of the structural and regulatory differences between the 3D architecture of the Drosophila genome and that of organisms lacking pairing; indeed, the close proximity of homologs may influence the manner in which loops and domains are formed in Drosophila in addition to playing a role in gene regulation^{7,74} (AlHaj Abed, Erceg, Goloborodko et al. bioRxiv³⁹).

Our observations have further revealed a correlation between pairing and Zld-mediated chromatin accessibility during zygotic genome activation. In this way, they champion a relationship between trans-homolog genome organization and key developmental decisions, aligning well with the growing recognition that pairing can play a potent role in gene regulation, even at some loci in mammals (reviewed by^{5,7}).

Minor comments:

1. Like in mammalian systems, on diagonal boxes of enriched contact frequency have been termed TADs in *Drosophila*. Why are the authors not considering the domains they observe, particularly the cis domains, TADs?

*We thank the Reviewer for pointing out that we did not make our use of the terminology clear. Since the formation and nature of “boxes” along the Hi-C diagonal in *Drosophila* is still a field of active research, there is currently no consensus in nomenclature. Thus, we have opted simply to use the generic term of “domains”. To avoid confusion, however, we have modified the text as follows:*

Page 6, line 135: When assembled into a map and analyzed at 1 kb resolution without regard to parental origin of the mapped fragments, these reads revealed features that are routinely seen in non-haplotype-resolved Hi-C maps^{42,44,45} – a central cis diagonal representing short-distance interactions in cis (cis read pairs) as well as signatures for compartments, domains (also called contact domains²⁶ and topologically associated domains, or TADs^{46,47}, hereafter referred to as ‘domains’), and interaction peaks – thus confirming the quality of our Hi-C data (Supplementary Fig. 2b). Presence of these interphase hallmarks indicates that the vast majority of the cells are in interphase rather than mitosis, where such features are absent⁴².

2. The authors analyze data from Zelda depleted embryos to understand the relationship between homolog pairing and Zelda-mediated chromatin opening. In Zelda-depleted embryos there is a decrease in boundary strength at regions of high PS and strong Zelda binding. Since such boundaries are also seen in cis are the Zelda-mediated changes more related to the cis domains, with the trans interactions a consequence or indirect effect of changes in cis?

This is a very exciting question. Unfortunately, however, because the data from Zld-depleted embryos were obtained from a Hi-C study that could not resolve haplotypes, they cannot be used to answer questions that require the distinction of cis from trans-homolog interactions. Nevertheless, with respect to the wild-type hybrid embryos, cis boundaries coincide with analogous trans-homolog boundaries, which may suggest the potential impact and relationship between the two. We have added the following new text to the manuscript:

Page 12, line 296: Thus, Zld-mediated chromatin accessibility and transcription appear correlated with the establishment of homolog pairing in the early embryo (Fig. 4e), reinforcing the notion that the establishment of homolog pairing may bear a relationship to genome function during key developmental events. For example, Zld-mediated opening of genomic regions may facilitate homologous sequences finding each other when pairing is initiated in early embryogenesis. Whether Zelda plays a direct or indirect role in initiating pairing, however, will require additional study, as will the interplay between Zld and other factors involved in pairing and other forms of chromosome organization.

3. The authors compare Fig. 3L to Fig. 3M. Fig. 3M is an observed/expected map, is Fig. 3L also an observed/expected map? If not, this comparison can't be made.

Good point! Since the expected (E) is the same for cis Pat and cis Mat in Fig. 3l, the ratio of two observed maps is mathematically equal to the ratio of observed/expected (O/E) maps. However, the Reviewer is correct in that, by neglecting to explain this simplification, our figure becomes more than confusing. We have renamed the header in Fig. 3l and S5f from 'cis Pat / cis Mat' into 'cis Pat O/E / cis Mat O/E' to more accurately describe the figure. Thank you!

4. The manner in which the section on mammalian homolog pairing was presented left the impression that no conclusions could be drawn in regards to mammalian homolog pairing so it seems like this discussion does not add much to the current manuscript.

We appreciate the Reviewer pointing this out and agree that our wording did not make clear our goals. In brief, a primary goal of the mammalian section is to highlight the generality of our haplotype-resolved approach, showcasing how it can be applied to any kind of inter-chromosomal contacts in any sequenced organism. Note, also, that we have now decided to be more circumspect in our conclusions. Thus, we point out that although some observed trans-homolog contacts may be suggestive of global transient pairing in mammalian embryogenesis, the sparsity of data preclude definitive conclusions, leaving this issue open for future studies. We have now modified this section accordingly:

*Page 13, line 306: One of the most exciting developments in recent years has been the growing indication that homolog pairing is not special to *Drosophila*, that it may be a general property in many organisms, including mammals (reviewed by^{5,7}). Thus, in order to both demonstrate the applicability of Ohm as well as assess the state of pairing in mammals, we applied our haplotype-resolved Ohm approach to mammalian embryos. As pairing in mammals is generally transient and localized, for example, during DNA repair, V(D)J recombination, imprinting, or X-inactivation (reviewed by^{5,7}), these analyses would, at the least, also serve as a control for observations in *Drosophila*, where pairing is far more extensive.*

Page 14, line 324: Requiring ≥ 2 SNVs per read, we turned to the datasets representing embryos ranging in age from the single, 2-, 4-, and 8-cell stage through the earliest days of embryogenesis and beyond. We observed no conclusive signal for trans-homolog contacts (Supplementary Figs. 9 and 10), consistent especially for the very earliest stages with previous indications that the maternal and paternal genomes are spatially segregated (^{32,33,62-64}; Supplementary Fig. 9). More specifically, although the signal for trans-homolog contacts for some of the earlier stages seemed to hover above that expected from homolog misassignment (Supplementary Fig. 9) and thus raise the possibility of homolog pairing, this discordance between observed and expected trans-homolog contacts could also reflect artifacts arising from a non-uniform genomic density of SNVs and/or Hi-C biases (see⁴⁸); we were unable to resolve this issue due to the sparsity of data (Supplementary Table 4). The data also raise the possibility of a higher order clustering of homologous as well as heterologous subtelomeric regions (Supplementary Fig. 10), reminiscent of the clustering of pericentromeric and subtelomeric regions seen previously⁶². While, again, the sparsity of data precluded our ability to determine such clustering reflected any degree of homolog pairing, it is worth noting that telomere as well as centromere clustering has been reported in embryos across several species^{15,44,62,65} and may constitute a mechanism for initiating pairing^{20-22,66}. In brief, although we found no definitive

evidence of homolog pairing in early mouse embryos, our data nevertheless leave this issue open for future studies. Also, as repetitive sequences were excluded from our analyses, the potential of trans-homolog contacts at repeats remains to be explored.

Page 16, line 379: *Our observations have further revealed a correlation between pairing and Zelda-mediated chromatin accessibility during zygotic genome activation. In this way, they champion a relationship between trans-homolog genome organization and key developmental decisions, aligning well with the growing recognition that pairing can play a potent role in gene regulation, even at some loci in mammals (reviewed by^{5,7}). As such, it may be of significance that homologs are unpaired during the earliest moments of Drosophila embryogenesis^{75,76} and then initiate pairing at a time coinciding with activation of the zygotic genome. This in mind, we examined data pertaining to early mouse embryo; as in Drosophila, murine maternal and paternal genomes are segregated in the earliest cell cycles^{32,33,62-64}, but little is known about the genome-wide status of pairing thereafter. Our analyses were further fueled by studies suggesting that pairing may be a general mechanism by which organisms detect structural heterozygosity in their genomes early in development, culling those that are deleteriously rearranged⁷⁷⁻⁸⁰ (bioRxiv⁸¹ and reviewed by⁷). Although we observed some intriguing signals, ultimately, the sparsity of data did not allow us to draw strong conclusions regarding pairing in the mouse embryos, pointing to a need for additional haplotype-resolved Hi-C analyses and/or the resolution provided by imaging.*

5. Perhaps the reviewer missed this, but are there trans-homolog compartments?

The Reviewer brings up a very interesting point. Thus, we have used Pearson's correlation analyses globally to check for trans-homolog compartments. Indeed, the trans-homolog compartments appear in early embryos, although they are weak. To this end, we have added additional Fig. S5a, and have adjusted the text accordingly:

Page 2, line 34: *This computational approach, which we call Ohm, revealed pairing to be surprisingly structured genome-wide, with trans-homolog domains, compartments, and interaction peaks, many coinciding with analogous cis features.*

Page 4, line 93: *Strikingly, our Ohm approach reveals that homolog pairing is extensive along entire lengths of chromosomes at short- as well as long-range distances and is highly structured, with trans-homolog domains, domain boundaries, compartments, and interaction peaks.*

Page 10, line 224: *We next asked whether our haplotype-resolved maps could further clarify the molecular structure of paired homologs as well as elucidate how trans-homolog interactions are integrated with the cis interactions that shape the 3D organization of the genome. Remarkably, we observed trans-homolog domains, domain boundaries, compartments,*

and interaction peaks resembling analogous features in cis maps (Fig. 3i-k and Supplementary Fig. 5a-d). In fact, 54% of the boundaries of trans-homolog domains overlapped a boundary of a cis domain (averaged over two homologs, Supplementary Fig. 5e).

*Page 15, line 357: Using Ohm, we obtained, for the first time, a genome-wide high-resolution view of homolog pairing during early *Drosophila* development. Our data revealed genome-wide juxtaposition of homologs along their entire lengths as well as a multi-layered organization of trans-homolog domains, domain boundaries, compartments, and interaction peaks. We also observed concordance of trans-homolog and cis- features, arguing that cis and trans interactions are structurally coordinated (see also AlHaj Abed, Erceg, Goloborodko et al. bioRxiv³⁹).*

We thank all Reviewers, again, for their supportive comments and helpful suggestions, which have guided us well in improving our manuscript.

Reviewers' comments:

Reviewer #1 (Remarks to the Author):

The authors have improved the manuscript by clarifying the difference in aims and focuses from the accompanying study. Authors also performed an additional analysis regarding the role of Zld and Bcd. However, data representation in new Figure S6 is not intuitively understandable, and thus the logic and argument in the text are not very clear for the readers. After improving this point, I think the paper will be ready for publication.

Reviewer #2 (Remarks to the Author):

This paper is technically sound and straightforward. I still think the main finding has been reported in their companion paper.

Reviewer #3 (Remarks to the Author):

I thank authors for thoroughly addressing my comments. I have two responses at this time, which, after being addressed, will make the manuscript suitable for publication in Nature Communications.

1. The additional figure the reviewers provided me in regard to trans-homolog interaction peaks (my major comment #2) should be included in the manuscript. If the authors are going to claim they detect trans-homolog interaction peaks in early embryos, it is important to show this data in the manuscript and not simply refer to their companion study for an additional discussion of loops/peaks as this study is on a cell line, not embryos.
2. Regarding my minor comment #3, the response by the authors states that the expected values for cis Pat and cis Mat are the same. Perhaps the reviewer missed this, but this should be shown, not simply claimed. Since the authors have enough SNVs and a robust computational approach, it is important to show this is the case.

Reviewers' comments:

Reviewer #1 (Remarks to the Author):

The authors have improved the manuscript by clarifying the difference in aims and focuses from the accompanying study. Authors also performed an additional analysis regarding the role of Zld and Bcd. However, data representation in new Figure S6 is not intuitively understandable, and thus the logic and argument in the text are not very clear for the readers. After improving this point, I think the paper will be ready for publication.

We appreciate the Reviewer pointing this out and agree that our wording did not make our findings clear. Thus, we have now made the following modifications to the text:

*Page 11, line 261 (in the main text): We correlated the binding of these factors, as revealed by ChIP-seq datasets^{52,54,57}, to local variations in the degree of pairing as characterized by the 'pairing score (PS)', defined as the log₂ trans-homolog contact frequency within a 28 kb window along the diagonal of Hi-C maps (Supplementary methods). Excitingly, regions associated with elevated PS values were significantly enriched for the binding of Zld and Bcd ($P < 10^{-10}$, Fig. 4a) as compared to controls in which we randomized 200 kb chunks of PS values (Fig. 4b) or 200 kb chunks of binding profiles (Supplementary Fig. 6a). No correlation was observed, however, between PS values and the binding of GAF or Dl ($P = 0.139$, $P = 0.029$, respectively; Fig. 4b). Then, using partial correlation analyses, we examined whether these correlations would be affected upon removal of the contribution of a third feature. For instance, how might the correlation between the binding of Bcd and PS values be affected by the binding of Zld? We found that Zld binding contributes significantly, as Spearman's correlation coefficient dropped from 0.155 ($P = 2.1 \times 10^{-65}$; Fig. 4b) to 0.044 ($P = 1.3 \times 10^{-6}$; Supplementary Fig. S6b), when controlled for Zld binding. Interestingly, Spearman's correlation coefficient between the binding of Dl and Ps values decreased from -0.020 ($P = 0.029$; Fig. 4b) to -0.157 ($P = 3.0 \times 10^{-67}$; Supplementary Fig. 6b) when controlled for the contribution of Zld. These findings are consistent with a role of Zld in Bcd- as well as Dl-dependent binding and gene activation^{52,54}. Interestingly, we observed that genes^{58,59} associated with both AP and DV patterning are similarly associated with higher PS values (Supplementary Fig. S6c). Moreover, loss of Bcd binding in Zld-depleted embryos⁵² was significantly enriched at regions otherwise associated with high PS values ($P = 1.21 \times 10^{-68}$; Supplementary Fig. S6d, e). **In sum**, our data suggest that Zld-mediated early opening of chromatin, including the binding of Bcd, is strongly correlated with pairing in embryos.*

Page 36, line 730 (in the Supplementary Fig. 6 legend): Supplementary Fig. 6. Zld-mediated opening of chromatin affects Bcd-dependent activity and its relation to pairing.

*Page 36, line 735 (in the Supplementary Fig. 6 legend): **b**, Partial Spearman correlation analyses showing, for instance, whether the correlation between a feature in a column (the PS or random PS values) and the binding of Zld (grey box on the right side) is affected after removal of any contribution of features displayed in the rows, such as the binding of Bcd, GAF, and Dl. Similar to assessments pertaining to Zld, partial correlation analyses were performed for each of the other factors highlighted in grey boxes. Partial Spearman correlation coefficients indicated in each box and by a heatmap; P-values in parentheses.*

Reviewer #2 (Remarks to the Author):

This paper is technically sound and straightforward. I still think the main finding has been reported in their companion paper.

Thank you very much for helping us improve the quality of our manuscript.

Reviewer #3 (Remarks to the Author):

I thank authors for thoroughly addressing my comments. I have two responses at this time, which, after being addressed, will make the manuscript suitable for publication in Nature Communications.

1. The additional figure the reviewers provided me in regard to trans-homolog interaction peaks (my major comment #2) should be included in the manuscript. If the authors are going to claim they detect trans-homolog interaction peaks in early embryos, it is important to show this data in the manuscript and not simply refer to their companion study for an additional discussion of loops/peaks as this study is on a cell line, not embryos.

We appreciate the Reviewer's encouragement to add examples of trans-homolog interaction peaks. Thus, we have now included them as Supplementary Fig. 5a. We have adjusted the main text and supplementary figure legend accordingly.

Page 10, line 227 (in the main text): Remarkably, we observed trans-homolog domains, domain boundaries, compartments, and interaction peaks resembling analogous features in cis maps (Fig. 3i-k and Supplementary Fig. 5a-e).

Page 35, line 709 (in the Supplementary Fig. 5 legend): a, Trans-homolog, cis maternal, and cis paternal maps showing interaction peaks on 2L and 2R.

2. Regarding my minor comment #3, the response by the authors states that the expected values for cis Pat and cis Mat are the same. Perhaps the reviewer missed this, but this should be shown, not simply claimed. Since the authors have enough SNVs and a robust computational approach, it is important to show this is the case.

The Reviewer raises a good point! We have now included Supplementary Fig. S5g to show that the expected values for cis maternal and cis paternal contacts are very similar.

Page 35, line 714 (in the Supplementary Fig. 5 legend): g, Contact frequency plotted against genome separation for chromosomes 2 and 3 normalized by the cis maternal contact frequency at 1 kb. Note that normalizing by the cis paternal contacts at 1 kb gave essentially the same result.

REVIEWERS' COMMENTS:

Reviewer #1 (Remarks to the Author):

I think the current form of manuscript is suitable for publication in Nature communication.

Reviewer #3 (Remarks to the Author):

All of my comments have been addressed and I thank the authors for doing so. The manuscript is now ready for publication in Nature Communications! Congratulations!